# Analytical and Numerical Study of Underwater Tether Cable Dynamics for Seabed Walking Robots Using Quasi-Static Approximation

Asghar Khan [1] , Xiangyu Wang [2,*], Zhenyu Li [1] , Liquan Wang [1], Ahsan Elahi [3] and Muhammad Imran [1]

1   College of Mechanical and Electrical Engineering, Harbin Engineering University, Harbin 150001, China; asghar81@hrbeu.edu.cn (A.K.); jade10623@sina.com (Z.L.); wangliquan@hrbeu.edu.cn (L.W.); mimran@hrbeu.edu.cn (M.I.)
2   Yantai Research Institute, Harbin Engineering University, Yantai 264006, China
3   College of Intelligent Systems Science and Engineering, Harbin Engineering University, Harbin 150001, China; ahsan.elahi@hrbeu.edu.cn
*   Correspondence: wangxiangyu325@126.com

**Abstract:** In this study, the dynamics of tether cables (TCs) that connect a surface ship and a walking robotic vehicle on the seabed are numerically investigated. The main aim of this study is to develop a reliable prediction model for the dynamic behavior of TCs attached to a seabed walking robot. This system consists of a surface ship, underwater manned seabed walking robot (UMSWR), TC, and winch. The study is comprised of mathematical modeling and numerical simulations of the developed governing equations for the TC dynamics. A relatively simple and efficient mathematical analysis method is proposed to determine the configuration and forces on the TC under a steady state. The problem is solved using the quasi-static technique and lumped mass parameters for discretizing and modeling the dynamics of the TC. Based on the static analysis of the Morrison equation and finite segment method, analytical formulas of the steady-state equation of TC were obtained and solved. The effects of variable water density and variable underwater currents are included in the cable behavior. Consequently, the two-dimensional TC profile and axial tension were estimated in a steady-state configuration. The developed equations were simulated in MATLAB software. Several numerical simulation examples were worked out, demonstrating the accurate performance of the method. Various input parameters of the system and their relationships with the output values were investigated, thereby demonstrating the versatility of the method.

**Keywords:** tether cable; finite element method; lumped mass parameter; axial tension; steady state; quasi-static

## 1. Introduction

The ocean is a significant source of food, energy, and valuable materials. In this regard, production from major offshore hydrocarbon fields began in the 1960s. The exhaustion of these fields and the continuous demand for hydrocarbon products has attracted offshore industry experts to explore and produce from fields located in deeper waters. These floating drilling/production systems are moored or tethered to the seabed via TCs [1]. For station-keeping seabed operating vehicles, continuous human presence in the loop necessitates the utilization of TCs. Knowledge of cable dynamics is very important in ensuring the underwater tethered platform responds properly to external disturbances. Generally, TCs are used for electrical power supply and communication purposes between the surface ship and underwater vehicle/system [2]. Tethers are a means for real-time telemetry and continuous power delivery between a surface station and a deployed underwater system. Moreover, TCs maintain a physical link with the deployed underwater vehicle and systems; therefore, they should work genuinely so that these systems operate safely and efficiently. It

is likely that TCs will be used extensively in future underwater vehicles and systems [3–6]. The success of an underwater tethered vehicle's operations is contingent on the ability of the human operator to control various operations of the vehicle. All the communications between the vehicle and the operator take place through the tether cable. To date, there are no wireless alternatives for subsea applications that can match the bandwidth provided by tether cables [7].

Tethered underwater vehicles equipped with robotic manipulators are used for many tasks, such as the extraction of specimens, the repair of underwater structures, and the inspection of subsurface phenomena. The success of these tethered vehicles greatly depends on the ability of the human pilot(s) to control the vehicle motion and the motion of the robotic manipulators through haptic and visual interfaces. A TC is a bundle of electrical conductors and fiber optics. Communication between the pilot and vehicle can occur in real time, and scientific data can also be transmitted to the surface through the TC. Moreover, the accuracy of the UV dynamic model and the navigation of the UV depends on minimal disturbances from the TC to some extent [8].

Numerous applications of marine cables, such as in oceanographic research, towing, hydrographic survey, salvage, telecommunications, and fishing, require accurate analysis to predict their static and dynamic behavior. A TC that connects a UV to the surface ship can be affected by many parameters, including the motions of either the UV or the vessel, the current along the cable, the total length of the cable itself, the diameter of the cable, and the density of the cable. TC configuration can be optimized through numerical simulations. TC analysis is a challenging task due to the difficulty in modeling the hydrodynamic forces, which in turn depend on the prevailing physical conditions. Hence, there is a need for a reliable and efficient solution technique for determining the cable profile and tensions [1,9–11].

Many researchers have modeled and analyzed underwater marine cable systems [6,11–15]. Generally, two types of techniques are used for tether cable analysis: steady-state analysis and time domain analysis. In [16], a finite element method was used to model the hydro-elastic instability of a ship-towed array system in axial flow. Computer codes were established, and several validation examples were carried out. In [17], Xu et al. used MATLAB code to model the position and tension estimation of an underwater cable system at certain depths using a finite element modeling and lumped mass technique.

There are several methods with different assumptions and considerations for the analysis of the motion of underwater cables or mooring lines, including analytical methods, the finite segment lumped mass method, the catenary method, the high-order finite element model, and the finite difference method. In the case of the high-order finite element analysis, there are more components and nodes as the length of the cable increases. The order of stiffness of the matrix rises as the number of elements and nodes increases, and the matrix becomes extremely sparse. Consequently, achieving convergence of the computing process is challenging and necessitates a significant amount of computation time. In the finite difference approach, discrete approximation of the differential, which is necessary for the finite difference technique, employs the function values of the discrete points to estimate the differential of the point. However, the finite difference technique is not appropriate for engineering issues with complex boundary conditions. An additional complication in applying the finite difference approach to the slack tether modeling is the limits on the mesh size that result due to the first-order finite differencing procedure. Due to these reasons, challenges exist in adapting both the finite difference and finite element cable modeling techniques to simulations of low-tension tethers. Of all these, the lumped mass method has been widely applied due to its ease of implementation and the accuracy of the numerical results.

The lumped mass and finite segment approach is the predominant method for submerged cable modeling and is considered to be more suitable for this work. As an element method, lumped parameter models are extremely modular and can easily accommodate the insertion of other elements such as vehicles, buoyancy/ depressor elements at interior

points, or end boundary locations of the discretization scheme. To date, the finite segment and lumped mass approach has remained extremely popular since its inception by Walton and Polacheck [18]. The problem can be applied to a variety of problems. It is used to study the response of buoy-suspended cables and pipelines in three dimensions to cross currents and wave motion. It can be used to study the dynamic analysis of single point moorings and multiple submerged buoys along a single mooring line, buoy-cable system for aquaculture application and multi-component mooring lines in the dynamic response of a station-keeping vessel. Moreover, tow cables and towed vehicles for various towing ship maneuvers, dynamics of a slender towed array of acoustic sensors in three dimensions and low-tension tether cables of ROVs can also be studied with this method. The flexibility of the method to simulate such a wide range of cases makes the lumped mass method a favored approach to offshore engineering experts.

In [19,20], the finite element modeling technique was used for the analysis of underwater cable systems. In these studies, underwater cables were studied as a series of rigid rods connected by frictionless joints. These segments have identical formulations of joints, forces, and constraints. The mass of each segment is supposed to be at the end of the segment under consideration. In other words, lumped masses are connected by inextensible massless rods. In this case, the effect of drag forces and fluid motion can be augmented in rods. Dynamic equations are iteratively solved using the boundary condition of surface ship position and velocity of the lower end. Buckham et al. [6] applied the finite element method to calculate the tension and bending force in a slack tether attached to an ROV.

In the quasi-static approximation method, the tether cable consists of a series of finite, linear, inextensible segments connected by massless pin joints. In this case, the mass of the tether segments is assumed to be distributed evenly over the length of the finite linear segments. In the finite segment and lumped parameter models, the tether elasticity is neglected, and the elements are considered to be rigid. The tension within each element is considered as an additional state variable, and geometric constraints that ensure constant element length are used to iteratively solve for these tension values. The assembly of forces that operate over the entire cable element are lumped at the center of gravity of the element. Many people have studied the quasi-static approach for the dynamic analysis of underwater cables. In [21], a quasi-static problem in two dimensions using a lumped parameter model explicitly defined the nodal velocities, which were integrated using a fourth-order Runge–Kutta integrator to produce a history of cable motion. In [22], an analysis of the quasi-static dynamics of a slender towed array of acoustic sensors in three dimensions was presented. In this study, tether cable was taken as a series of finite, linear, inextensible segments. The mass of the tether segments is to be distributed evenly over the length of the finite linear segments, giving rise to the inclusion of rotational inertia terms and coordinates specifically to the orientation of the tether segments. In [23], finite segment formulation against experimental data describing the motion of an anchor cable was validated by Kamma et al. They also presented the use of this finite segment model in simulating buoy motion in three dimensions [24].

The use of numerical models to simulate the dynamics of underwater vehicles and marine cables has been presented extensively in the literature. Numerical models provide a means for the analysis of TC dynamics, deep water moorings, and marine risers before the actual development and/or deployment of the system [25]. An application that has not yet seen significant attention is the simulation of underwater seabed walking vehicles with slack tethers. Despite advances in autonomous technology, underwater tethered vehicles are still the predominant tool for complex intervention tasks. These vehicles typically follow omnidirectional paths during operation and deploy low-tension or slack neutrally buoyant tethers along the path. Travel to the limits of the tether, sudden movements of the UVs and/or environmental loads can cause the tether to become taut. To develop numerical dynamics models for the motion of the slack TC, it is necessary to develop an accurate representation of the external and internal effects that contribute significantly to

the motion of the slackened tether. To ensure that the tether cables forces transmitted to the UV are minimal, the tether cable design is such that it retains relatively high flexibility and neutral buoyancy. In addition, an extra tether cable is deployed, creating a curvilinear tether lay, to ensure a low-tension or slack state [11].

In [26], the dynamic behavior of a marine cable with variable length during turning maneuvers were explored. The cable was discretized into a lumped mass model using the lumped mass method, and a dynamic analysis model of the turning process of the marine cable was established for different release speeds and water depths. Time domain coupling analysis was used to determine the dynamic changes in configuration and stress of marine towing cable at various speeds and depths of water. In [27], the finite element model of the variable-length underwater cable with geometrically nonlinear motion was presented. Nonlinear, time-varying differential equations are derived from the theorem of linear and angular momentum, and its weak formulation was obtained by the principle of D'Alembert–Lagrange. Via a series of consistent linearization with the Frechet derivative and discretization with the iso-parametric interpolation, the governing finite element model for this variable-length cable was established. Then, the corresponding Newmark implicit time integration formulation was derived, and a scheme of adaptive step size was proposed to avoid convergence difficulty during the numerical calculation. The performance of the proposed approaches is further assessed with numerical cases, which take into consideration the top alternating excitation, the terminal follower forces, and the sea current along the cable. In [28], a study on the dynamic modeling and the motion simulation of an unmanned ocean platform, an unmanned underwater vehicle with an underwater cable, was performed. Newton's second law and lumped mass method were used to derive the equations of motion of unmanned surface vehicles, unmanned underwater vehicles, and underwater cables.

In [29], modeling the behavior of an underwater vehicle operating in the coastal water, including the tether cable effect, was presented. A new simulation method for combining the rigid body motion of the underwater vehicle and the flexible motion of the cable equation was performed. For flexible cable dynamics modeling, governing equations of the TC dynamics were established based on the catenary equation method, and the shooting method was applied to solve the two-point boundary value problem of the catenary equation. The formulation and solution of the governing equations to estimate the position and forces of the tether cable endpoint under the action of concentrated and distributed forces due to underwater currents were proposed. In [30], motion analysis of a coupled unmanned surface vehicle, umbilical cable and unmanned underwater vehicle system was investigated, in which multi-body dynamics of the coupled system was employed. The unmanned surface vehicle and unmanned underwater vehicle are modeled as rigid-body vehicles, and the flexible cable is discretized using the catenary equation. For the nonlinear coupled dynamics of the vehicles and flexible cable, the fourth-order Runge–Kutta numerical method is implemented. For cable dynamics, the shooting method is applied to solve a two-point boundary value problem of the catenary equation. A computer simulation was used to study the behavior of the coupled unmanned surface vehicle, umbilical cable, and unmanned underwater vehicle system. Resultantly, the variations in the cable forces and moments at the tow points and the underwater configuration of the cable were investigated.

According to the work presented in [31], the onboard operator will control vehicle locomotion and manipulator operation through real-time telemetry and visual interfaces afforded by TC. The study of tether cable dynamics being deployed from surface ship to seabed walking robot, an instance of slack tether in seabed operations, was a primary motivation for the furthering of the finite segment and lumped mass approach. In this case, TC generally maintains a low-tension state. Environmental disturbances that accumulate over the TC may affect UMSWR operations to some extent. The focus of this work is the development of a low-tension TC dynamics model for application in UMSWR simulation. In this work, the development and implementation of a lumped mass strategy for the

dynamic modeling of TC for UMSWR is carried out. The theoretical and mathematical development of the model will be discussed. It includes a discretization scheme for TC geometry and the methodology for the calculation of the internal and external forces acting over the TC. The modeling methodology presented provides the foundation for the subsequent developments of strategy for the simulation of the TC motion that occurs during the locomotion of UMSWR on the seabed. In simulating the hydrodynamic performance of the TC system, the coupling effect between the TC and the UMSWR is neglected, and the hydrodynamic model is composed of TC dynamics only. For applications where knowledge of the bending and torsional moments are not considered, linear elements are used since the linear lumped mass elements do not exhibit any curvature or twist. While it does not include low-tension bending and torsional effects, the lumped mass and finite segment approach has been demonstrated accurately in underwater hydrodynamic applications while requiring significantly less computational expenses than the similar finite segment approach. Given the success of simple linear models, we choose to follow the approach of these simpler methods to the approximation of the hydrodynamics, tether profile and axial forces acting over the tether element.

The second most important motivation and challenge is the inclusion of the variable environmental parameters in the modeling of TC dynamics. Generally, in underwater cable systems, one of the key factors affecting system analysis accuracy is the influence of the underwater environment on the behavior of the cable system. The most important environmental parameters include variable water density and variable surface currents. This variation in the seawater density results in the net positive buoyancy or net negative buoyancy force on the cable in water. This buoyancy effect must be included for hundreds and thousands of meters of submerged cable. Similarly, variable surface current changes the position and posture of the flexible underwater cable very easily, which results in a change in the position and posture of the underwater cable system. After conducting a literature survey of existing work on underwater cable problems, it was revealed that many researchers had dealt with cable problems, neglecting the variable surface current and variable water density effects. Considering the environmental parameters in underwater cable problem analysis had been treated rarely by a few. Water current is either neglected or taken as a constant value. For water density, it is taken as a constant value, but in practice, the density of water changes with depth. Anticipating deficiency with these features, the author decided to develop a new formulation, which could readily handle the inclusion of variable environmental parameters, and which could easily be implemented on physical systems.

The remainder of this manuscript is organized as follows: Section 2 describes the problem statement and the system components used in this study. Included in the section is a description of the tether cable attached to the seabed walking robot and associated surface platform arrangement, and information about the tether cable. Section 3 discusses the methods used in modeling tether cable dynamics. First, an explanation of the basic assumption, mathematical modeling, reference frame system, environmental parameters, cable discretization, and element force definition is given. Next, the development of governing equations of motion for the tether cable procedure is discussed, followed by tether cable geometry and tension computation and the steps necessary for numerical calculations and numerical implementation of the dynamic model. Section 4 presents the results of the simulation study. Figures are presented that show the tether cable profile and tension values in the tether cable and their dependence on various input parameters for the simulation cases that were performed. Calculations are mentioned and explained in context with their effects on the results. Then, the results of the cable profile and tension values in the tether cable are discussed. Section 5 summarizes the study and formulates conclusions based on the results obtained in Section 4. Authors' contribution, Funding, Conflict of interest and References follow the conclusion at the end of the paper.

## 2. Problem Statement

The system for the study consists of a surface ship, Underwater Manned Seabed Walking Robot (UMSWR), TC and winch, as shown in Figure 1. UMSWR is used for seabed operations of pick and place, inspection, repair, and accumulating seabed sampling applications. UMSWR walks on the seabed in a straight line with an average speed of 1 m/s or 2 Knots.

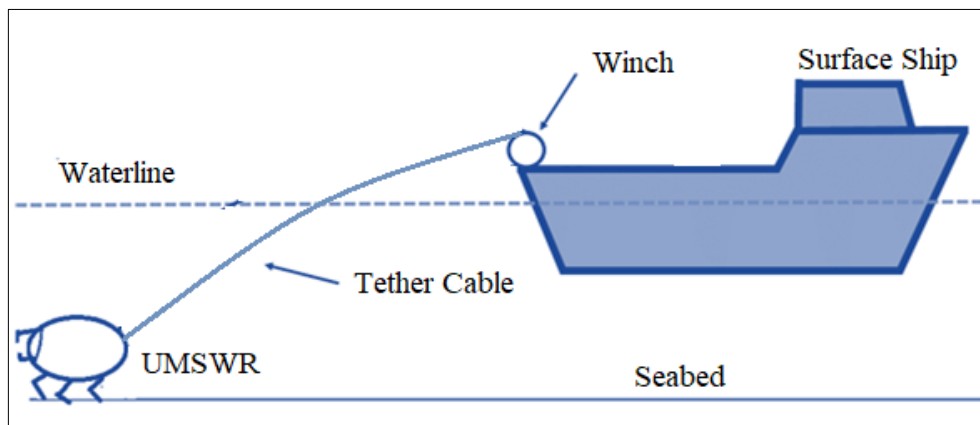

**Figure 1.** UMSWR connected to surface ship via tether cable that walks on seabed.

For all its operations and locomotion on the seabed, power is obtained from the surface ship via TC. Moreover, all sort of communication between the surface ship and onboard operator takes place through TC, which contains fiber optic cable. The upper end of the TC is connected to the surface ship, which is fixed on the water surface using a dynamic positioning control mechanism [32]. The lower end of TC is connected to the UMSWR that walks on the seabed with constant velocity. TC moves along with UMSWR, the tangential velocity of TC being the same as that of the UMSWR. However, the velocities of the TC elements in the x-axis direction vary linearly from bottom to top due to the curved shape of the cable profile. The steady-state analysis of the tether cable is performed for various input parameters. Another motivation for this work is how to minimize tether-induced vehicle disturbances and correctly simulate the low-tension tether dynamics. There have been many efforts to minimize tether-induced vehicle disturbances. One of these methods to reduce these disturbances includes the control of the tether payout. This causes the tether cable length to be a variable parameter. Numerous studies have been conducted on variable-length TCs for towed underwater vehicles and cable laying applications [5,9,12,33]. Through simulation of the pay-in and payout of low-tension TCs, active control of the tether length may help to minimize the tether-induced disturbances at the vehicle. Incorporating a variable length capability in the study will overcome such issues in the modeling of low-tension cable dynamics.

The steady state analysis of the tether cable is performed using lumped mass and finite segment approach using quasi-static approximation. In the present study, a tether cable is adopted as the numerical model for calculations. The TC is divided into 200 equal-length elements, and the result from one element is propagated into the next till it reaches the tow point at UMSWR. Two types of frames of reference are used in the study, and these are the inertial frame of the reference and the cable-fixed frame of reference. The inertial reference frame (X, Z) is defined at the surface of the waterline, and the first cable's element is attached to the surface ship. The cable used has a neutral buoyancy of 1025 kg/m$^3$, variable water density having a linear profile and variable water current having an exponential profile are considered. The position, posture, tether profile and tension force in the tether cable are determined for various input parameters. Moreover, the effects of various input parameters and their relationships with the output values are discussed in detail.

## 3. Materials and Methods

### 3.1. Modeling of the Tether Cable System

In [34], tether cable selection from electrical and mechanical perspectives is discussed. For the design of tether cable, primary inputs such as the vehicle missions, number of major loads and maximum power demand are considered. Moreover, the maximum depth of operation, and data related to the weather and water conditions at the operating site should be collected such as wave height, current, wind speed and water temperature. For subsea applications, generally, multicore cables are used due to their compactness and minimizing the number of connectors. A generalized cable is depicted in Figure 2.

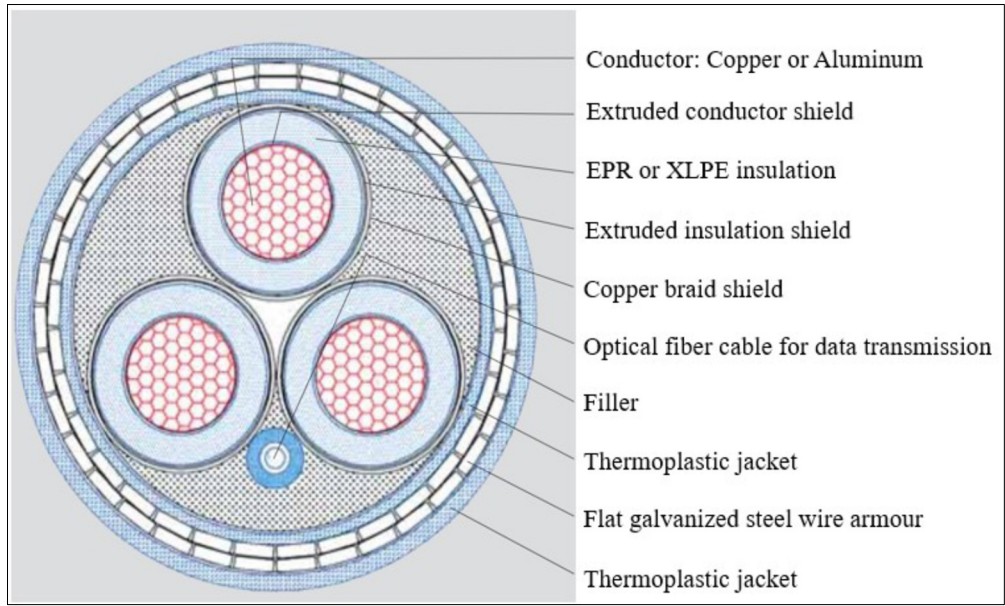

**Figure 2.** Cross-section of a typical three-core tether cable with double-layer armoring.

In this paper, we present the derivation and development of governing equation of TC motion for a finite-segment lumped parameter model. The model developed is limited to a two-dimensional (2-D) configuration, and hence, 2-D analysis. This study is conducted to estimate the profile and forces on underwater tether cables. The effects of variable water density and variable underwater currents are included in the cable behavior. Water density variation occurs due to temperature and salinity changes in various areas. Marine water density usually varies from 1025 kg/m$^3$ to 980 kg/m$^3$ up to 500 m depth of water. The current variation occurs with the depth of water and becomes almost zero at a depth of 50–60 m below the water surface [10]. This modeling approach will enable us to observe, predict and monitor the TC dynamic response in any operational conditions. The study comprises the mathematical modeling and numerical simulation of the developed governing equations for the TC under consideration. Consequently, the 2-D TC profile and axial tension are estimated in a steady-state configuration.

### 3.2. Basic Assumptions

- TC is continuous, inextensible, and infinitely flexible.
- TC can only resist tension forces, not bending moment and compressive forces.
- Hydrodynamic forces on TC are resolved into tangential and normal components.
- Steady static dynamic condition is considered, and the problem is addressed at equilibrium condition.
- Length of the TC is not fixed and can be adjusted as the robot moves forward or backward using payout or pay-in operations.
- Underwater current is parallel to the UMSWR heading and varies vertically with depth, and becomes zero at a depth of 50–60 m below the water surface.

- Seawater has variable water density.
- UMSWR is heading in a fixed straight line and with a nominal constant speed.
- No surface and subsurface disturbances for surface platforms are considered.

*3.3. Mathematical Modeling*

The mathematical model of the system is useful not only for formulating control algorithms but can also be used for performing the simulation. A continuous cable model cannot be used for the solution of the problem; it is, therefore, more appropriate to use a discrete model of the cable. Several approaches have been used by various people to model tether cable dynamics. There are several methods with different assumptions and considerations for the analysis of underwater cables. These include the analytical method, finite segment lumped mass method, catenary method, high order finite element model and finite difference method. Of all methods in the literature, the lumped mass method and finite segment method have been widely applied due to their ease of implementation and accuracy in terms of numerical results [35–38]. We employed lumped parameter and finite segment methods for this tether cable study. In this method, TC is discretized into a finite number of inextensible rectilinear segments. Each segment has an identical formulation of parts, joint forces and constraints. These segments are connected by frictionless pin joints at the node position. The use of pin joints makes the TC very flexible. Hydrodynamic, gravitational, and buoyant forces and the mass of the cable are concentrated at the center of mass of the cable elements. The TC is divided into $N$ equal-length elements, and the result from one element is propagated into the next till it reaches the endpoint of the TC. The number of discrete elements defines the accuracy of the simulation. The greater the value of $N$, the greater will be the accuracy. For $N$ cable elements, there are $N + 1$ nodes. An illustration of the discrete representation of the marine cable is shown in Figure 3. This implies that the TC is modeled by an assembly of $N$ rectilinear elements that extend between node 1 and node ($N + 1$), the boundary nodes. Node 1 lies at the point of connection of the TC with the surface ship, while node ($N + 1$) lies at the point of connection of TC with the UMSWR.

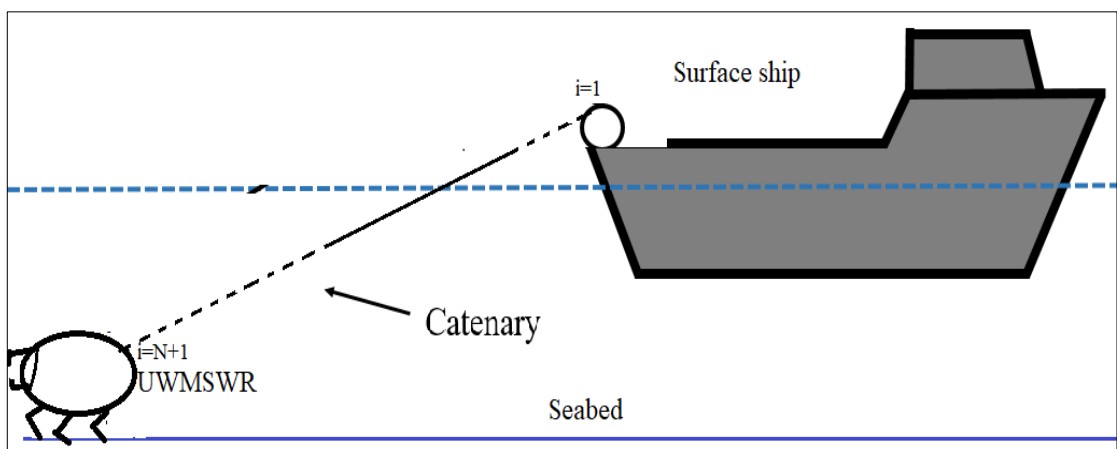

**Figure 3.** Discretization of the TC into N segments.

This problem is based on a two-dimensional formulation in which the steady motion of the UMSWR in a straight path is considered. Hence, using a quasi-static approximation for TC dynamics for modeling TC motion and resulting cable profile and axial tension. For the relatively steady motion of the marine tethered platform, the transient motions of the cable during straight maneuvers are not significant. Thus, the problem is formulated as a simple quasi-static problem in which the inertial terms, including added mass effects in the cable equations of motion, are ignored, and the cable is assumed to respond instantly to any external disturbances. Inertial forces and added mass effects are therefore neglected in this study. Using a quasi-static approximation to the tether dynamics, the equilibrium

states of a tether cable were determined during straight motion and at the constant speed of the seabed walking robot. The simulation is extended for various input parameters. The basic input parameters to the solution of the problem are UMSWR speed, operating depth, cable diameter, number of cable segments, TC density, length of the TC, seawater density and underwater current. In the output, our parameters of interest include the position and velocity of each node, depth, altitude, and velocity of each segment and ultimately, the axial tension along the cable and its profile. Such modeling provides numerically efficient simulation without the loss of accuracy. Since these linear lumped mass elements do not exhibit any curvature or twisting phenomena, it does not include low-tension bending and torsional effects. The lumped mass and finite segment approach has been demonstrated accurately in hydrodynamic applications while requiring significantly less computational expense than the other numerical modeling approach. On the other hand, for example, the finite difference modeling approach is based and built on a more mathematical foundation involving complex mathematical computations.

*3.4. Reference Frame System*

Coordinate reference frames are necessary to describe any position or motion of the system and its components [8,28]. The use of additional reference frames makes the derivation of the equations of motions easier. While using multiple reference frames, there is one important issue related to the transformation of vector coordinates from one frame to another. For this purpose, rotation matrices are used based on the Euler angles. A reference frame is used to determine distance and direction in the system. A coordinate system is used to represent measurements in a frame [39]. In the case of the 2-D model, two types of orthogonal reference frames are employed. These frames are inertial reference frame $(i, k)$, with its origin fixed in space and TC body-fixed reference frame $(t, n)$. These reference frames are illustrated in Figure 4.

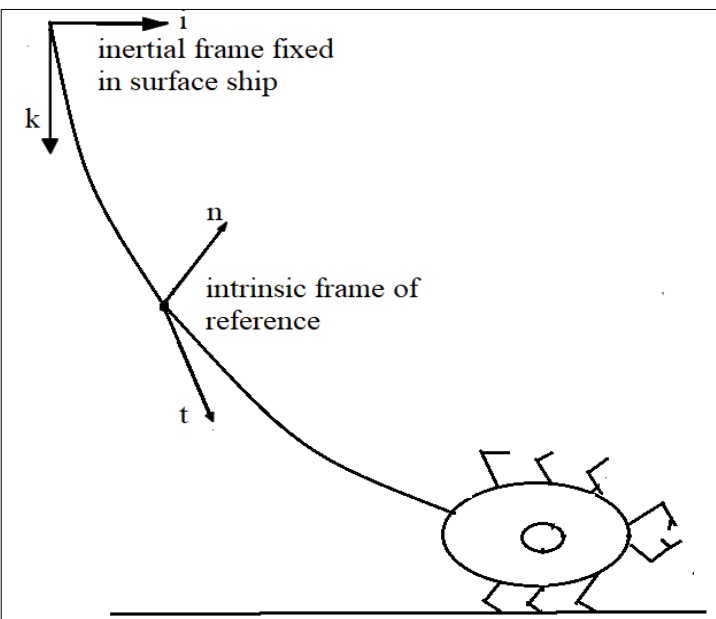

**Figure 4.** Inertial and cable–body fixed reference frame definition.

Important dynamic properties of the TC are analyzed in a better way in Cable-Fixed Frame while the governing equations of TC motion representation are required with respect to the inertial reference frame. For actual cable, the orientation of the local reference frame in two-dimensional space varies continuously with changing location both in space and time, as depicted in Figure 5. This frame is also called the moving frame, as the direction of the unit vectors changes for each element in time. The origin of the local frame of reference is placed at the center of gravity of the element. The inertial frame of reference has its origin

placed at the point where the cable departs from the surface support ship, and the first cable's element is attached to the surface support ship. The unit vector $i$ is in the surface ship longitudinal direction with $k$ pointing vertically downwards. The surface vessel is assumed to be fixed in position and, hence, the origin of the inertial reference frame.

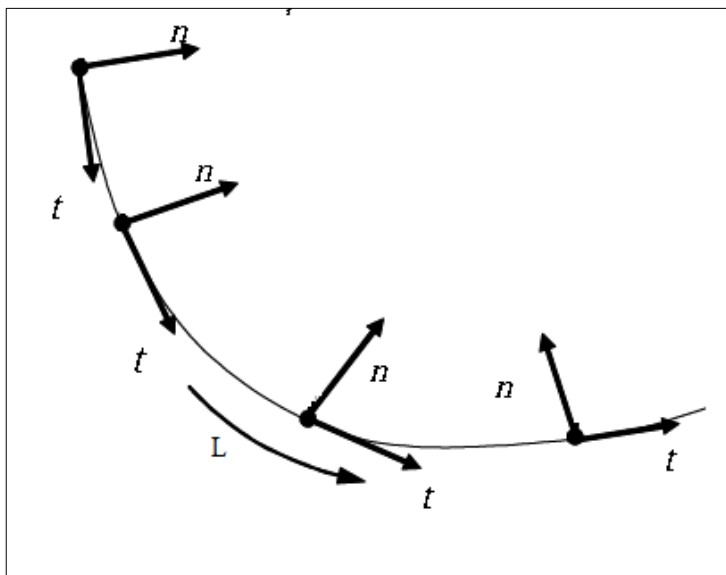

**Figure 5.** The local frame moving along the TC changing directions of its unit vectors continuously.

The equations of motion of the tether cable must be transformed into an inertial frame of reference. Equations of motion are transformed from intrinsic frame of reference $(t, n)$ to an inertial reference system with unit vector $(i, k)$. This transformation occurs by using a set of Euler angles. For the 2-D case, the Euler angles are limited to a single rotation $\theta$. We use a transformation matrix $R_{IB}$ that can be obtained through a partial rotation, which is used to relate two two-dimensional axes systems. $R_{IB}$ can be defined as:

$$R_{IB} = \begin{bmatrix} \cos(\theta) & -\sin(\theta) \\ \sin(\theta) & \cos(\theta) \end{bmatrix}, \tag{1}$$

for each element of the TC, the inertial frame $(i, k)$ and TC body fixed frame $(t, n)$ can be related by using Equation (2):

$$\begin{bmatrix} i \\ k \end{bmatrix} = R_{IB} \begin{bmatrix} t \\ n \end{bmatrix}. \tag{2}$$

The inverse relation can be obtained as:

$$\begin{bmatrix} t \\ n \end{bmatrix} = R_{BI} \begin{bmatrix} i \\ k \end{bmatrix} \text{ whereas } R_{BI} = R_{BI}^{-1}.$$

The Euler angles can be calculated at any instant in the simulation, provided that the endpoints of the cable element are known.

Key factors affecting TC analysis accuracy are the influences of underwater environment parameters. They change the position and posture of the flexible cable, which results in the variation of loads on underwater vehicles. In order to ensure the required accuracy, these factors should be included in the analysis. The most important of these factors include variable water density and variable surface current [40]. These environment parameters must be specified in a mathematically precise way.

### 3.5. Variable Underwater Current

Variable underwater current effects are included in TC analysis. Ocean currents are continuous, predictable, directional movement of seawater, driven by gravity, wind friction

(Coriolis Effect), and water density variation. It is primarily horizontal water movement, ranging in magnitude from 0.02 Knots to as much as 5 Knots [41]. The ocean is constantly in motion and never stands still. Hydrodynamic forces accumulate on TC, either due to underwater currents or relative motion of the UMSWR and surface ship, which may affect its maneuverability. Underwater currents generally may change direction and speed with depth. Underwater currents are expressed as a function of water depth $z$:

$$v_c = f(z). \tag{3}$$

Usually, these currents become equal to zero at a depth of 50–60 m below the surface of the water. In scientific study, various underwater current profiles are considered, including constant profile, linear profile, exponential profile, and sinusoidal profile. Figure 6 shows linear and exponential profiles of the underwater current with respect to depth. The most realistic profile for engineering applications is the exponential profile. Actual measurements at sea can also be used to calculate the profile of the underwater current for that area. The relative velocity $V$ of the fluid around the tether cable can be expressed as relative velocity due to UMSWR $v_r$ and underwater current velocity $v_c$:

$$V = v_c + v_r. \tag{4}$$

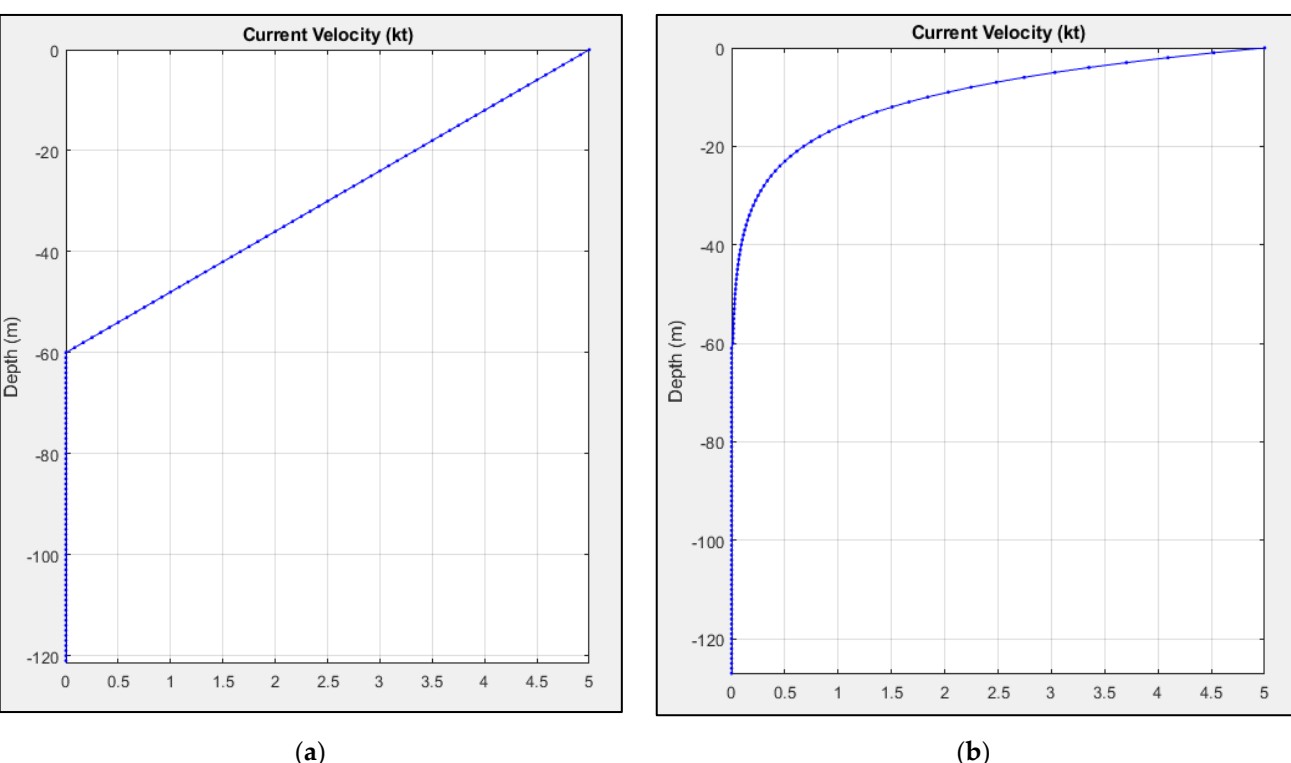

(**a**)                                                             (**b**)

**Figure 6.** Underwater Current Profiles: Linear profile (**a**) Exponential profile (**b**).

As UMSWR moves on the seabed with velocity $v_r$, the end node $(i = N + 1)$ of TC moves at the same speed, i.e., $v_r$. The position of the first node of the TC that leaves the winch is fixed in space and has a velocity equal to zero. All elements of the TC move at a speed, uniformly varying from $v_{N+1,r} = 1 \times v_r = v_r$ at node $(i = N + 1)$ to $v_{1,r} = 0 \times v_r = 0$ at node $(i = 1)$, which results in a uniform velocity gradient of the TC element in the x-axis direction. This velocity gradient is then augmented with the surface water current $(v_c)$ that results in the relative velocity of the TC and surrounding fluid medium.

$$V_i = v_{i,r} + v_c. \tag{5}$$

The relative velocity of the fluid $V_i$ is a function of depth z. This means that each element of the cable will have a distinct drag velocity at different depth locations.

### 3.6. Variable Water Density

Different marine waters in the world have different densities that primarily depend on water temperature and salinity [42–44]. This variation in the water density results in the net positive buoyancy or net negative buoyancy force on the object in the water. Similarly, this buoyancy effect has to be studied for hundreds and thousands of meters of tether cable. Variable water density $\rho_0$ is a function of water depth z from the surface of the water:

$$\rho_0 = f(z). \tag{6}$$

Generally, water density varies linearly with depth z, ranging from 1025 kg/m$^3$ on the water surface to 990 kg/m$^3$ at a depth of approximately 500 m from the surface [10]. This effect is shown in Figure 7. This density variation with depth has a significant effect on the tether cable's specific weight and, hence, buoyancy force, on the tether cable.

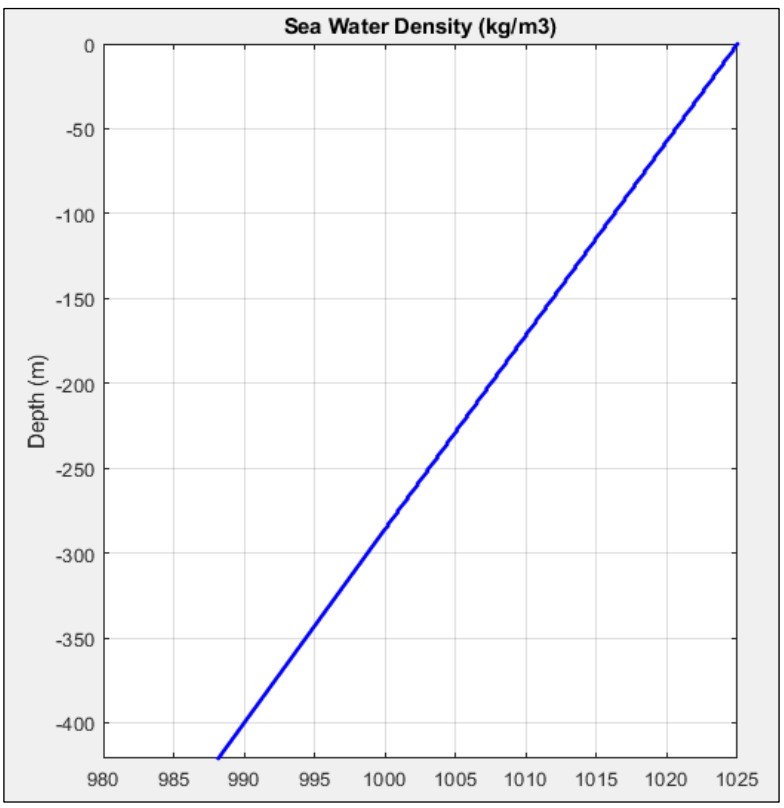

**Figure 7.** Linear Seawater Density Profile vs. depth.

### 3.7. Cable Discretization

The whole cable of length L is divided into N number of segments. The length of each segment l can be calculated as follows:

$$l = L/N. \tag{7}$$

For N segments of the tether cable, there are (N + 1) cable nodes. These cable nodes start at node $i = 1$ at the point where the cable is connected to the surface ship/floating platform where the winch of the cable is located. Its number increases from top to bottom, with Node N + 1 at the point where the cable is connected to UMSWR.

### 3.8. Element Force Definition

For the cable element, the discretization and force diagram are illustrated in Figure 8. The schematic shows various internal and external forces acting on the cable element *i*. The external forces include weight force $W_i$, buoyancy force $B_i$, axial drag force $D_{a,i}$ and normal drag force $D_{n,i}$. Hydrodynamic and gravitational forces acting over the TC create internal forces, i.e., axial tension $T_i$ within the cable elements, which significantly disturb the motion of tethers and, hence, UVs [11]. Internal forces in TC are generated by the elasticity of the cable elements, which allow the stretch of the elements in the tangent direction. Huang [45] has proposed a methodology for calculating cable tension in three dimensions. With this approach, the tension $T_i$ within the element *i* acts in the tangential direction of the element, modeled as a linear function of the tangential strain of the discrete cable elements. However, in slack or low-tension cables, the stretch is so small that it is neglected. The tension in such a case is found by an assembly of forces in the element by using steady-state conditions in equilibrium. Generally, fundamental dynamic properties and forces on the cable are analyzed in the cable fixed frame and are then transformed into the inertial reference frame [46]. The weight force $W_i$ of the cable element *i* can be expressed as:

$$W_i = \left(\frac{\pi d^2}{4}\right) l\rho g, \tag{8}$$

where $d$ is cable diameter, $\rho$ is cable density, $l$ is length of cable element, and $g$ is the gravitational acceleration constant and $g = 9.8\,\text{m/s}^2$.

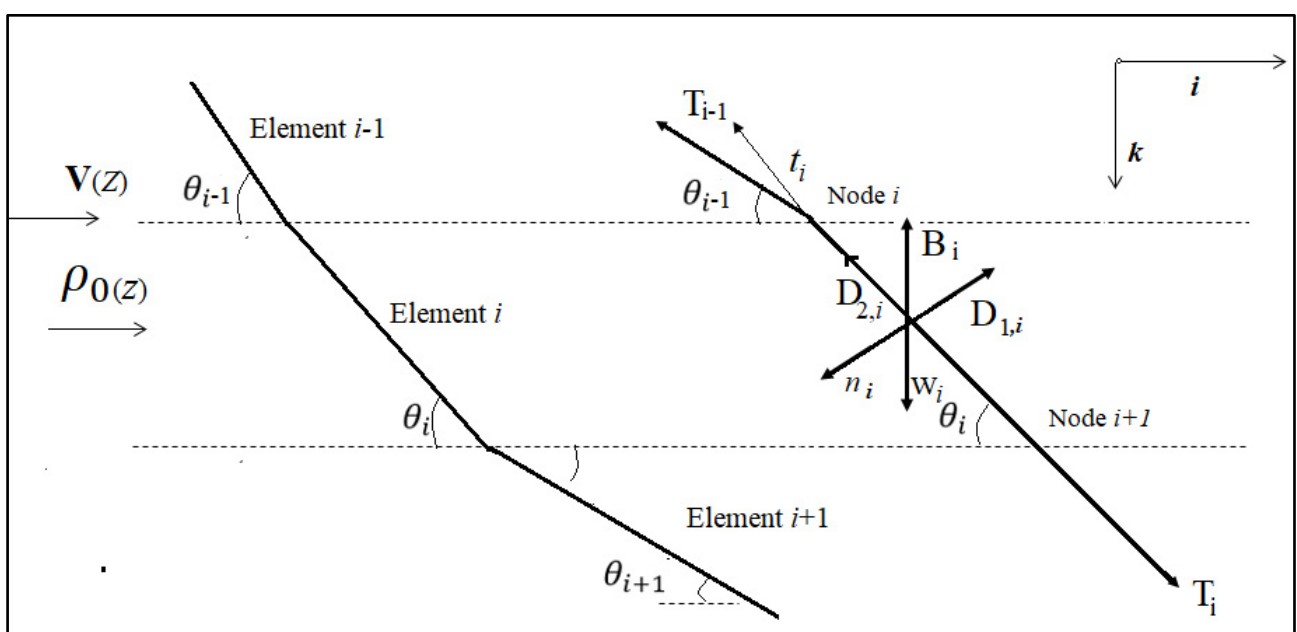

**Figure 8.** TC element model and Force schematic.

Similarly, buoyancy force on the cable element *i* is calculated as Equation (9):

$$B_i = \left(\frac{\pi d^2}{4}\right) l\rho_0 g. \tag{9}$$

Here, $\rho_0$ represents water density. Consequently, specific weight of the cable $W_{s,i}$ is expressed as shown in Equation (10):

$$W_{s,i} = W_i - B_i. \tag{10}$$

$W_{s,i}$ is the specific weight of the *ith* cable element at the geometric center of the element with respect to the depth of the element. $W_{s,i}$ varies with depth as the density of water $\rho_0$ changes with depth. So, the density of the element is taken at the average depth of the element by linear interpolation of the end nodes' Z coordinates ($\frac{Z_i+Z_{i+1}}{2}$), which is mathematically the mid-point of the element. Putting values of $W_i$ and $B_i$ into Equation (10), the specific weight of the element can be written as in Equation (11):

$$W_{s,i} = \left(\frac{\pi d^2}{4}\right) lg(\rho - \rho_0), \tag{11}$$

the drag force acting on the cable is divided into tangential and normal components. According to hydrodynamic Morison formula, the axial drag force on element $i$ is $D_{a,i}$ which acts parallel to the t-axis of the cable, i.e., tangent to the cable element. $D_{a,i}$ is given as shown in Equation (12):

$$D_{a,i} = \frac{1}{2} C_{da} \rho_0 \pi \, d \, l \, V_i^2 cos^2 \theta_i, \tag{12}$$

where $C_{da}$ is the coefficient of the axial drag of the cable, $V_i$ is the relative velocity at depth of the element, and $\theta_i$ is the angle of the element, correspondingly.

Relative velocity $V_i$ depends on the depth of water. $V_i$ is the velocity of the geometric center of the *ith* cable element with respect to the surrounding fluid. $V_i$ is found at the average depth of the element by linear interpolation of the end nodes' Z coordinates, i.e., ($\frac{Z_i+Z_{i+1}}{2}$), also called the midpoint height of the element, as shown in Figure 9.

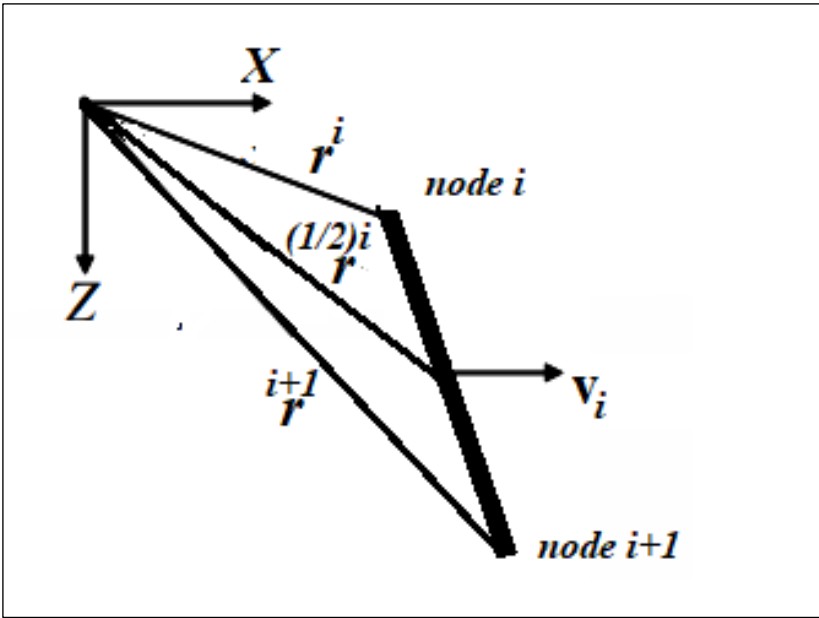

**Figure 9.** *ith* element estimation of the velocity based on element's geometric center.

By taking the constant terms ($\frac{1}{2}C_{da}\rho_0 \pi \, d \, l$) equivalent to $D_1$ for simplification, Equation (12) can be written:

$$D_{a,i} = D_1 V_i^2 cos^2 \theta_i. \tag{13}$$

Similarly, the normal or cross-flow drag force $D_{n,i}$ is the force that acts normally to element $i$. It acts along the n-axis of the cable. $D_{n,i}$ is calculated by Equation (14):

$$D_{n,i} = \frac{1}{2} C_{dn} \rho_0 d \, l \, V_i^2 sin^2 \theta_i. \tag{14}$$

where $C_{dn}$ is the coefficient of the normal drag of the cable. To simplify the equations, taking the constant terms $\frac{1}{2}C_{dn}\rho_0 d\ l = D_2$, Equation (14) can be written as:

$$D_{n,i} = D_2 V_i^2 sin^2\theta_i. \tag{15}$$

For *N* cable elements, there are *N* angles formed, one each at the lower end of every element. For element *i*, the angle is $\theta_i$ at node $(i+1)$. Tension $T_i$ of the element *i* makes an angle of $\theta_i$ at node $(i+1)$ and the tension $T_{i-1}$ at the previous end of the same element makes angle $\theta_{i-1}$, which acts in the opposite direction.

It is obvious that all the forces ($W_i$, $B_i$, $D_{a,i}$, $D_{n,i}$ and $T_i$) are a function of the water depth because the relative velocity $V_i$ and water density $\rho_0$ depends on the water depth.

*3.9. Assembly of Forces*

Applying the assembly of forces at the lumped mass representation of the cable elements yields a series of 2N equations that govern the motion of the element nodes in the X and Z directions. To evaluate such equation of motion, the vector quantities described above are resolved into inertial representation for use in the equations of motion of the cable element. By applying the assumption of steady-state condition at equilibrium, as mentioned earlier, the force balance for the *ith* element in the cable-fixed frame can be expressed as shown in Equation (16):

$$R_{IB,i}\sum F_i + R_{IB,i-1}\sum F_{i-1} + W_{s,i} = 0. \tag{16}$$

Equation (16) can be written as:

$$R_{IB,i}\left(\begin{bmatrix} -T_i \\ 0 \end{bmatrix} + \begin{bmatrix} D_{a,i} \\ 0 \end{bmatrix} + \begin{bmatrix} 0 \\ D_{n,i} \end{bmatrix}\right) + R_{IB,i-1}\begin{bmatrix} T_{i-1} \\ 0 \end{bmatrix} + \begin{bmatrix} 0 \\ W_{s,i} \end{bmatrix} = 0. \tag{17}$$

Putting the values of the relevant transformation matrix in Equation (17), we obtain the system of Equation (18) for element *i*:

$$\left.\begin{array}{c} -T_i cos\theta_i + D_{1,i}V_i^2 cos^3\theta_i + D_{2,i}V_i^2 sin^3\theta_i + T_{i-1}cos\theta_{i-1} = 0 \\ and \\ T_i sin\theta_i - D_{1,i}V_i^2 cos^2\theta_i sin\theta_i + D_{2,i}V_i^2 sin^2\theta_i cos\theta_i - T_{i-1}sin\theta_{i-1} - W_{s,i} = 0 \end{array}\right]. \tag{18}$$

We obtain (2*N*) the number of equations for *N* elements, which govern the motion of the nodes in the X, Z directions. Equations of system (18) for element *i* depend on the terms from the previous element $(i-1)$. These terms can be written as:

$$C_1 = T_{i-1}cos\theta_{i-1}. \tag{19}$$

$$C_2 = -T_{i-1}sin\theta_{i-1}. \tag{20}$$

Putting the terms containing quantities from the previous elements, the system of Equation (18) becomes:

$$\left.\begin{array}{c} -T_i cos\theta_i + D_1 V_i^2 cos^3\theta_i + D_{2,i}V_i^2 sin^3\theta_i + C_1 = 0 \\ and \\ T_i sin\theta_i - D_{1,i}V_i^2 cos^2\theta_i sin\theta_i + D_{2,i}V_i^2 sin^2\theta_i cos\theta_i + C_2 - W_{s,i} = 0 \end{array}\right]. \tag{21}$$

The equations of system (21) represent the governing equations of TC dynamics for UMSWR in 2-D. Equation (21) presents two unknowns $T_i$ and $\theta_i$ for each element. In order to solve these equations, an iteration solution must be used along with boundary

conditions. For $i = 1, 2 \ldots N$ and constraints $\left( \theta_i \neq 0 \text{ and } \theta_i \neq \frac{\pi}{2} \right)$ the iterative solution to Equation (21) gives solution for $\theta_i$ as:

$$D_{1,i}V_i^2 tan\theta_i sin\theta_i + C_1 tan\theta_i + C_2 - W_{s,i} = 0. \tag{22}$$

And for axial tension $T_i$ as:

$$T_i = \frac{D_{1,i}V_i^2 sin^3\theta_i + D_{2,i}V_i^2 cos^3\theta_i + C_1}{cos\theta_i}. \tag{23}$$

The main concern for the tether cable analysis is the axial force or tension, which is the most crucial component of studying cables. The ultimate objective is the determination of the tension in each element of the TC as a function of the relative velocity $V_i$ and, consequently, the tension of TC at the point where it is connected to UMSWR.

*3.10. Geometry Computation*

The orientation of each discrete cable element is represented by using a $\theta_i(X_i, Z_i)$ Euler angle set [45]. These successive rotations align the inertial frame with the *ith* local frame of the cable. To specify a particular orientation for the local frame, all values for angle ($\theta_i$) are calculated based on the observed lay of the linear element in space. From Figure 8, element *i* that is bound by two nodes *i* and *i* + 1, we have:

$$R_{IB,i}\begin{bmatrix} 0 \\ l_i \end{bmatrix} = \begin{bmatrix} X_{i+1} - X_i \\ Z_{i+1} - Z_i \end{bmatrix}. \tag{24}$$

where $X_{i+1}$ and $X_i$ represent the location of nodes *i* and *i* + 1 of the element *i* in x-axis and while $Z_{i+1}$ and $Z_i$ represent location of the nodes of element *i* in z-axis Substituting the values of $R_{IB,i}$ in Equation (24) we get the following relation:

$$\begin{aligned} lcos\theta_i &= (X_{i+1} - X_i) \\ -lsin\theta_i &= (Z_{i+1} - Z_i) \end{aligned} \Bigg]. \tag{25}$$

From equation set (25)m each node coordinates of the cable *i* in both x-axis and z-axis can be calculated given the location of the previous node values:

$$X_{i+1} = (X_i + lcos\theta_i). \tag{26}$$

$$Z_{i+1} = (Z_i - lsin\theta_i). \tag{27}$$

From Equations (22), (26) and (27), after knowing the angles and coordinates of all the nodes for each element, we can get the cable profile. The horizontal distance between TC lower end and surface ship is called the trail or range of the TC. Similarly, the range of the TC is calculated using Equation (28):

$$Range = \sum_{i=1}^{N} l \times cos(\theta_i). \tag{28}$$

Similarly, the depth of the cable can be calculated using Equation (29), as below:

$$Depth = \sum_{1=1}^{N} l \times sin(\theta_i). \tag{29}$$

*3.11. Variable-Length Tether Cable*

To minimize tether-induced vehicle disturbances, the length of the TC should be controlled by the mechanism of tether payout and reel-in. This causes the TC length to be a variable parameter. Moreover, the hydrodynamic drag force acting on the cable induces a

large variation in the cable tension as the UMSWR moves forward. The difference between the amount of cable actually deployed and the cable required from the surface ship to reach the UMSWR point of connection is called slack. Careful control of the slack in the TC is extremely important in order to avoid the extremes of running out of cable, leading to extra cable on the seabed that results in zero tension and avoiding the taut conditions that build up excessive tension, which can break the cable. Therefore, the length of the cable should be compensated by either reeling in or paying out [5]. With this compensation mechanism, the winch rotation controls TC variation by either paying out the cable when the tension is increasing or reeling in the cable when the tension is decreasing. Paying out and reeling in the cable is carried out using a linear engine or a drum at a rate governed by the UMSWR speed of advance along the route.

The cable payout rate $V_{po}$ should be equal to the UMSWR speed to avoid any unfavorable conditions, i.e.,

$$V_{po} = v_r. \tag{30}$$

The amount of cable paid out is obtained through the integration of the payout rate. The payout rate is assumed to be a known function of time. Thus:

$$L_{po} = \int_t^{t+\Delta t} V_{po}(t)dt. \tag{31}$$

In practice, the amount of cable paid out $L_{po}$ is the length of extra cable added to the initial value of the cable necessary to control the slack. As the UMSWR goes forward in time interval $\Delta t$, $L_{po}$ is given by Equation (32):

$$L_{po} = v_r \times \Delta t. \tag{32}$$

By using the continuity relation, which states that the tether cable suspended length at time $(t + \Delta t)$ is the cable suspended length at time $t$ plus the amount of cable paid out $L_{po}$ during time interval $\Delta t$. Hence,

$$L = L_1 + L_{po}, \tag{33}$$

where $L$ is the total length and $L_1$ is the initial length at time $t$.

### 3.12. Numerical Calculation

To evaluate the motion of the TC and calculate its profile and tension in steady-state condition, the concept of state vectors is used. Because the state-vector technique accurately represents most physical systems, which only have very minor deviations in their steady-state behaviors. To obtain the solution, it is convenient to describe the system in terms of its intermediate dynamical state, called state vectors [47]. It expresses the effect of external forces on the system in quantitative form before the present moment so that the future evolution of the system can be exactly given from the knowledge of the present state and the future inputs. Consequently, the state vector can be used for state determination [48]. Hence, equations of motion for any physical system may be conveniently formulated in terms of its state for a given instant of time. Future states depend only on the current state and on any inputs at a given instant of time and beyond. All past states and the entire input history are summarized by the current state [49].

The steady-state state vectors (X, Z, θ) define the cable profile at any instant in the simulation and the state vector T defines the tension in the TC. Initially, state vectors of X, Z, T and θ are defined using zero vector and discretization parameter N for initialization of these parameters. These state vectors represent column vectors with N rows for T and θ and (N + 1) rows for X and Z, respectively. Given the initial values of the tether state vectors, the simulation can be advanced in a steady-state condition following these steps:

- First of all, using zero vector and discretization parameter *N*, initialization of the state vectors for X, Z, T and θ, which are used to allocate indices to node and element values for the state of these quantities given by:

$$X = \text{zeros } (N+1, 1), Z = \text{zeros } (N+1, 1), T = \text{zeros}(N,1), \theta = \text{zeros}(N,1). \qquad (34)$$

- Initial values of X, Z are given by X(1) = 0 and Z(1) = 0 by fixing the position of node $i = 1$. The value of T at element $i = 1$ can be found by using Equation (11), and $\theta$ at node $i = 1$ is assumed to be $\theta(1) = 30°$ for the initialization of these variables.
- Positions of X(2) and Z(2) for node $i = 2$ are calculated by using Equations (26) and (27).
- From these findings, the orientation of each element is calculated, and the series of rotational transforms is carried out by using Equation (21) at equilibrium conditions for all elements.
- The X and Z coordinates for all nodes $(2 < i \leq N + 1)$ were calculated by Equations (26) and (27).
- After finding the coordinates of all nodes for elements of TC, velocities are found for the average depth of the element, which is actually the velocity of the center of mass of the element.
- Using the known element velocities $V_i|_i^N$, the hydrodynamic drag forces in normal and tangential directions are calculated using Equations (13) and (15).
- Similar to element velocities, densities are calculated at the average depth of the elements of TC.
- Similarly, using the known values of the densities at the depth of the element, consequently, the specific weight $W_{s,i}|_i^N$ are calculated using Equation (11).
- Using the assembly of force at equilibrium condition at the center of mass of the element, the tension forces $T_i$ within all elements are calculated according to Equation (23). The tension at element $i = N$, is the tension at the tow point of the UMSWR.
- The element tension ($T_i$) is augmented with the orientation ($\theta_i$) of the current tether state to form the first steady state of the state vector for cable length $L_1$. This vector is then passed onward to produce the next state of the cable for cable length $L_2$. As the speed of the UMSWR is assumed uniform, the process continues for the limit of the tether cable maximum length.

Assembly of the forces and solution of the governing equations and implementation of the state vector technique is carried out in MATLAB-R2022b software on the latest desktop computer to simulate the governing equations of TC dynamics. The specifications of the desktop computer are: Core i-7, 2.60 GHz. The equations developed above are coded in a MATLAB program, and provisions are made to allow for steady-state motion, gravity loading, and velocity-squared dependent drag forces. MATLAB is used because of the programming simplicity and powerful matrix and vector operators in it, which significantly simplify the formulation and solution of the TC equations of motion [50].

*3.13. Boundary Conditions*

The *N* instances of Equation (16) form a combination of initial and boundary value problems. To evaluate the cable dynamics, it is necessary to provide both the requisite boundary conditions and an initial system state from which the model can be advanced in time. Boundary conditions must be given at both ends of the cable (upper end and lower end). Dirichlet boundary conditions are used in such kinds of simulations [51]. Since nodes of interest are located on the boundary, the Dirichlet boundary condition is straightforwardly applied to end nodes of the TC. For the nodal system depicted in Figure 3, the boundary conditions may be written as:

$$\begin{aligned} v_1 &= 0 \quad \text{at } (x = 0, y = 0), \\ T_1 &= W_s \quad \text{at } (x = 0, \ y = 0). \end{aligned} \qquad (35)$$

This gives boundary condition at the upper end of the TC. $v_1$ is the velocity of node 1 which is fixed in space. As we assume that the vessel motion is fixed by using dynamic position control, the first boundary condition is given by fixing the position of the surface

vessel. Whereas $T_1$ is the tension experienced at the upper end of the TC, which can be calculated by using the formula in Equation (36):

$$W_s = \sum_{i=1}^{N} W_{s,i} = \sum_{i=1}^{N} W_i - \sum_{i=1}^{N} B_i. \tag{36}$$

where $W_s$ gives the effective specific weight of the cable of a given length, which in turn gives the tension of the TC at the upper end. Similarly, for the lower end, the cable is connected to the UMSWR. So, the second boundary condition of TC is given as:

$$\begin{aligned} v_{N+1} &= v_r & \text{at } (x = X_L, \, z = Z_L), \\ T_N &= T_{Resultant} & \text{at } (x = X_L, \, y = Y_L). \end{aligned} \tag{37}$$

$v_{N+1}$ is the velocity of the lower end node that is connected to the UMSWR, which is equal to the velocity of the UMSWR itself. $T_N$ is the tension in the element $(i = N)$, which is equal to the resultant tension as the UMSWR moves at a given speed on the seabed for a given length L. $x_L$ and $z_L$ are the locations of the positions of end nodes on the cable attached to UMSWR on the seabed.

In the present study, a cable is adopted as the numerical model for calculations. TC is divided into 200 equal-length elements, and the result from one element is propagated into the next till it reaches the final element at the UMSWR. The global coordinates of the initial position of the TC connected at the free surface to the support vessel are assumed to be $(0, 0)$ m and at the other end connected to UMSWR are $(x_L, z_L)$m, respectively. The initial length of the TC is set to be 1000 m. The lower end of the cable is moving at the speed of the UMSWR $(v_r)$; the variable surface current speed $v_c$ is considered in the study. Initially, prior to movement of the UMSWR, the maximum tension is experienced at the upper end of the tether cable. As the UMSWR starts moving, the maximum tension is experienced at the lower end of the TC due to the hydrodynamic drag force.

*3.14. Implementation of Numerical Model and Simulation of Tether Cable Maneuvers*

The foregoing procedure has been developed into an algorithm for MATLAB computer programs. Using the 2-D lumped mass and finite-segment model formulation presented above, the motion of deployed TC subject to general kinematic or dynamic boundary conditions is captured. The performance of the lumped mass and finite-segment model for various TC parameters and the speed of the UMSWR are examined. The numerical implementation of governing equations of TC dynamics is carried out by the algorithm of Figure 10. This algorithm mainly focuses on the solver of Equations (22) and (23) for all cable elements and iterative update of the environment parameters and ensures that suitable environment parameters are applied to discrete cable elements at corresponding depth. Equation (22) is a continuous function having a unique zero in the interval $\theta_i = -\frac{\pi}{2}, \frac{\pi}{2}$. The stability of zero finding for Equation (22) relies on Parameter N.

*3.15. Tether Cable Effects*

While moving on the seabed, the maximum tension in the tether cable is generated at the tow point. The tension of the TC at the tow point, i.e., $T_c = T_N = (T_X, T_Z)$ results in the additional forces and moments that affects the motion of the UMSWR. In terms of the position of the tow point in the UMSWR, the cable-induced moments are calculated as

$$M_c = r_c \times T_c. \tag{38}$$

where $r_c = (r_X, r_Z)$ is vector from the center of the gravity of UMSWR to the connected point between the cable and UMSWR.

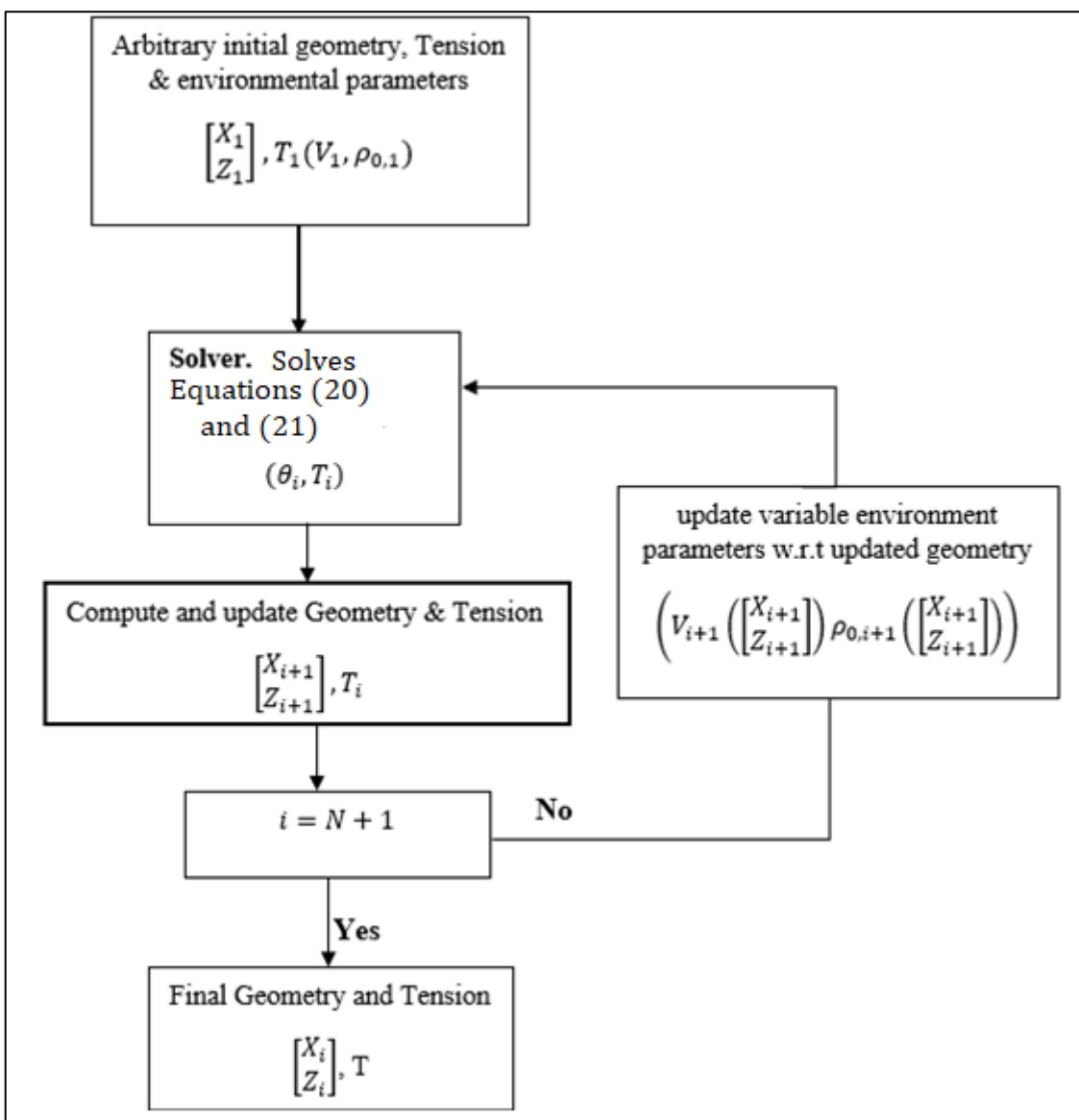

**Figure 10.** Schematic of numerical algorithm.

## 4. Results and Discussion

As the nonlinear dynamic equations are complex and difficult to solve analytically, the numerical simulation approach is used to simulate the motion of the TC. The developed mathematical model of TC dynamics is simulated by running certain simulation examples. The steady-state analysis of TC dynamics is performed for various input parameters of TC. Variable surface current and variable water density effects are included in the analysis in a mathematically precise way. Axial tension and TC profiles are calculated for various input parameters, and the relationship of the input parameters with the output value is analyzed and discussed.

### 4.1. Inputs

The input parameters include water current $v_c$, the density of the water $\rho_0$, the density of the cable $\rho_c$, water depth, cable discretization parameter $N$, cable length, cable drag coefficients ($C_{da}, C_{dn}$), and UMSWR speed $v_r$. These input parameters are summarized in Table 1.

**Table 1.** Typical numerical input parameters values for simulation.

| S. No | Parameter 2 | Value (Units) |
|:---:|:---:|:---:|
| 1 | Water density ($\rho_0$) (Linear Profile) | 1025 kg/m$^3$ |
| 2 | Gravity acceleration (g) | 9.8 m/s$^2$ |
| 3 | Cable Discretization ($N$) | 200 |
| 4 | Cable diameter ($d$) | 0.02 m |
| 5 | Cable density ($\rho$) | 1025 kg/m$^3$ |
| 6 | Cable cross drag coefficient ($C_{da}$) | 0.015 |
| 7 | Cable normal drag coefficient ($C_{dn}$) | 1.0 |
| 8 | UMSWR speed ($v_r$) | 2 Knots |
| 9 | Water Surface Current ($v_c$) (Exponential profile) | 5 Knots |

*4.2. Outputs*

Outputs of the simulation are TC configuration, tension profile and hence the maximum tension in TC. Various combinations of input parameters are investigated for these output values. The output and its relationship with various input parameters are discussed in detail.

*4.3. Variable Water Current Effects on TC Dynamics*

Analysis of selected tether cable parameters (Table 1) was performed at UMSWR constant speed of 2 Knots and seabed depth of 500 m for variable surface current values. The translational motion of UMSWR was restricted in the global X-direction. Each cable was discretized in 200 equal cable segments. The total force, position and orientation of each cable are obtained from the simulation. The effect of surface current on TC profile and tension are presented in Figures 11–14 for surface current values of 0, 1, 3 and 5 Knots, respectively. The effects of the simulation are summarized in Table 2. It is obvious from these figures that the surface current has a direct relation with the length of the cable length required to arrive at the seabed for steady motion of the UMSWR and TC, and consequently, the tension developed in the TC. The greater the value of the surface current, the greater will be its effect on TC tension and profile.

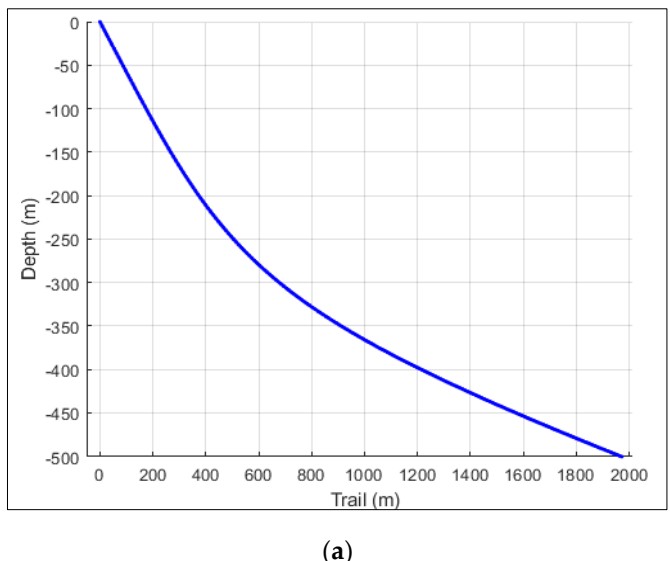

(**a**)

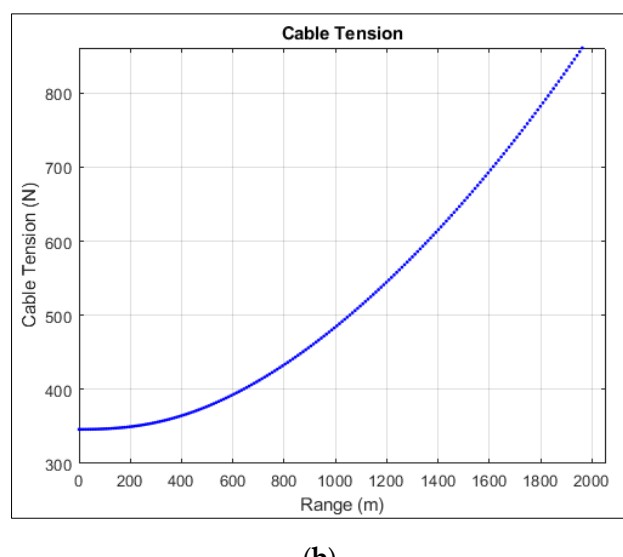

(**b**)

**Figure 11.** TC profile and tension estimated for zero surface current: (**a**) TC profile and (**b**) TC Tension.

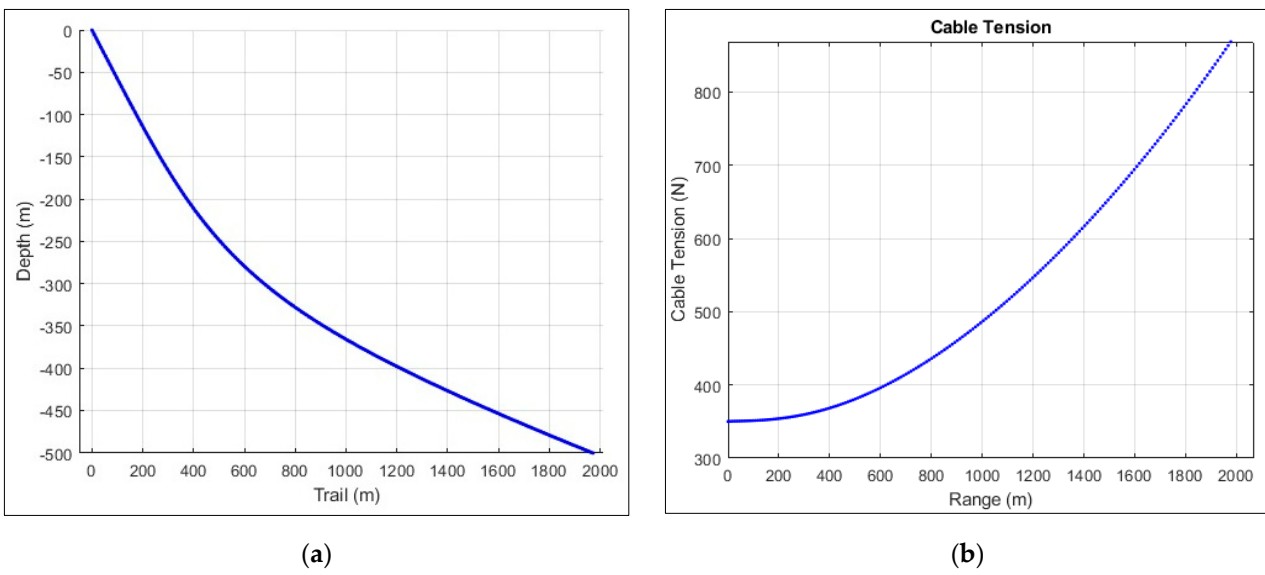

**Figure 12.** TC profile and tension estimated for zero Surface current of 1 Knots: (**a**) TC profile and (**b**) TC Tension.

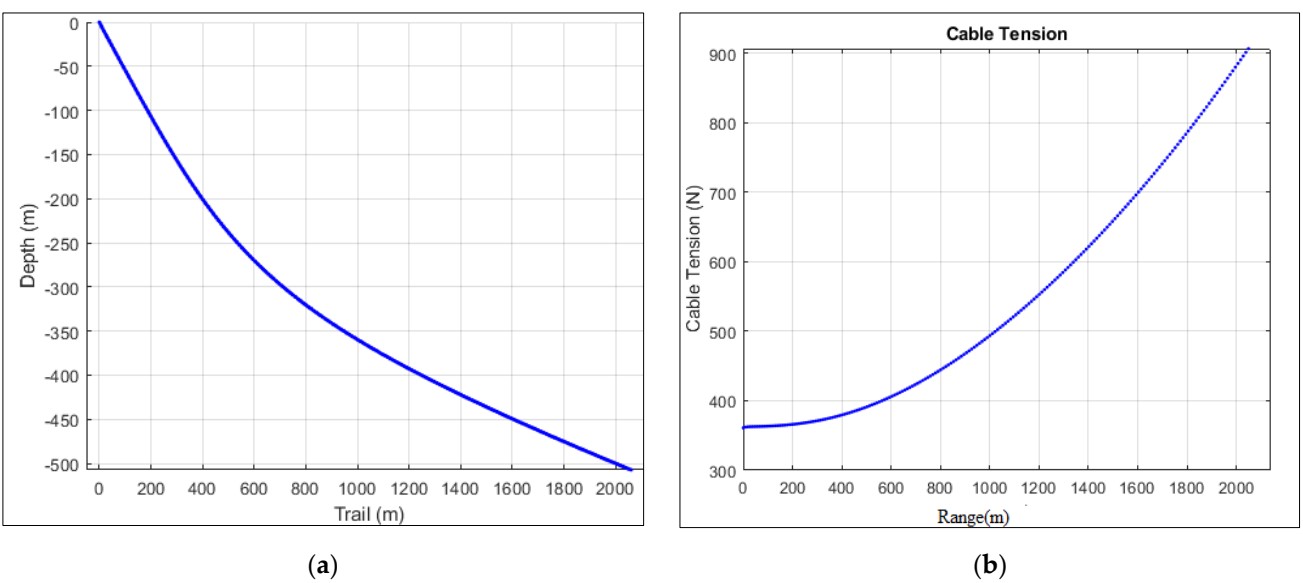

**Figure 13.** TC profile and tension estimated for zero surface current of 3 Knots: (**a**) TC profile and (**b**) TC Tension.

**Table 2.** Summary of the effect of the surface current on TC profile and tension.

| S. No | Surface Current (Knots) | TC Length Deployed (m) | Tension (N) |
|-------|-------------------------|------------------------|-------------|
| 1 | 0 | 2050 | 858 |
| 2 | 1 | 2070 | 877 |
| 3 | 3 | 2140 | 886 |
| 4 | 5 | 2260 | 928 |

It is due to the fact that tether cables are very flexible and very easily displaced by water currents. The greater the water current value, the greater will be the relative speed at the depth of the element of the cable. Consequently, the drag force on the cable elements increases, which results in greater tension values and larger cable length for the attainment of steady-state conditions of tether cable. Therefore, it is suggested that current data of the

area of operation must be properly collected and included in the simulation of the tether cable dynamics and for practical applications as well.

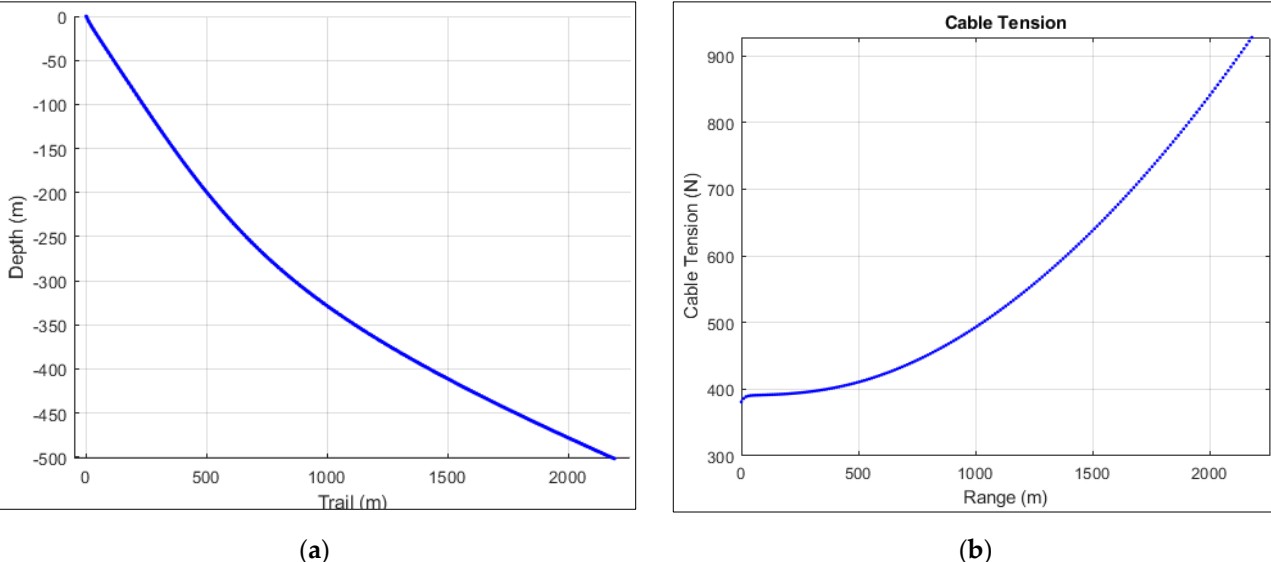

(**a**)                           (**b**)

**Figure 14.** TC profile and tension estimated for zero surface current of 5 Knots: (**a**) TC profile and (**b**) TC Tension.

### 4.4. Variable Water Density Effects on TC Dynamics

The effects of density on tether cable profile and tension are presented in Figures 15 and 16 for surface current values of 5 Knots. Cable density is taken as $\rho = 1050 \text{ kg/m}^3$. Two cases are considered, one for the constant value of water density $1025 \text{ kg/m}^3$ and the other for a variable water density having a linear profile. In the second case, water density varies from $1025 \text{ kg/m}^3$ on the surface to $980 \text{ kg/m}^3$ to a depth of 500 m. The effects of the simulation are summarized in Table 3. It is obvious from these figures that the density profile has a big effect on the length of the cable required to arrive at the seabed for steady motion of the UMSWR and TC and, consequently, the tension developed in the TC. In the case of a linear profile, the value of density decreases as we go deeper into the sea, resulting in lesser buoyancy force on the cable, making the TC heavier, and ultimately in the quick reach of TC to the depth in the study. It is obvious from the study that actual density values and their profile at the locality are very important as they have an impact on the TC length required.

### 4.5. UMSWR Speed Effects on TC Dynamics

The effects of UMSWR speed on tether cable profile and tension are presented in Figures 17 and 18 for surface current values of 5 Knot. Cable density is taken as $\rho = 1025 \text{ kg/m}^3$. Two cases are considered, one for 2 Knot speed of UMSWR and the other for 1 Knot speed of UMSWR. The effects of the simulation are summarized in Table 4. It is obvious from the simulation results that vehicle speed has a big effect on the profile of TC and tension developed in it. It is due to the high value of the relative velocity of the TC for the higher speed of the UMSWR as compared to a lower speed. The tension produced is the function of the relative velocity squared, as obvious from Equation (23). The higher value of the tension developed needs a bigger value of TC length to arrive at the seabed for the steady motion of TC. As the speed of UMSWR is increased, TC tension and, consequently, the TC length needed for arriving at a given depth of operation is increased accordingly. An increase in UMSWR speed, from 1 Knot to 2 Knot, results in a 31% increase in the cable length required and a 120% increase in the tension of TC at the tow point.

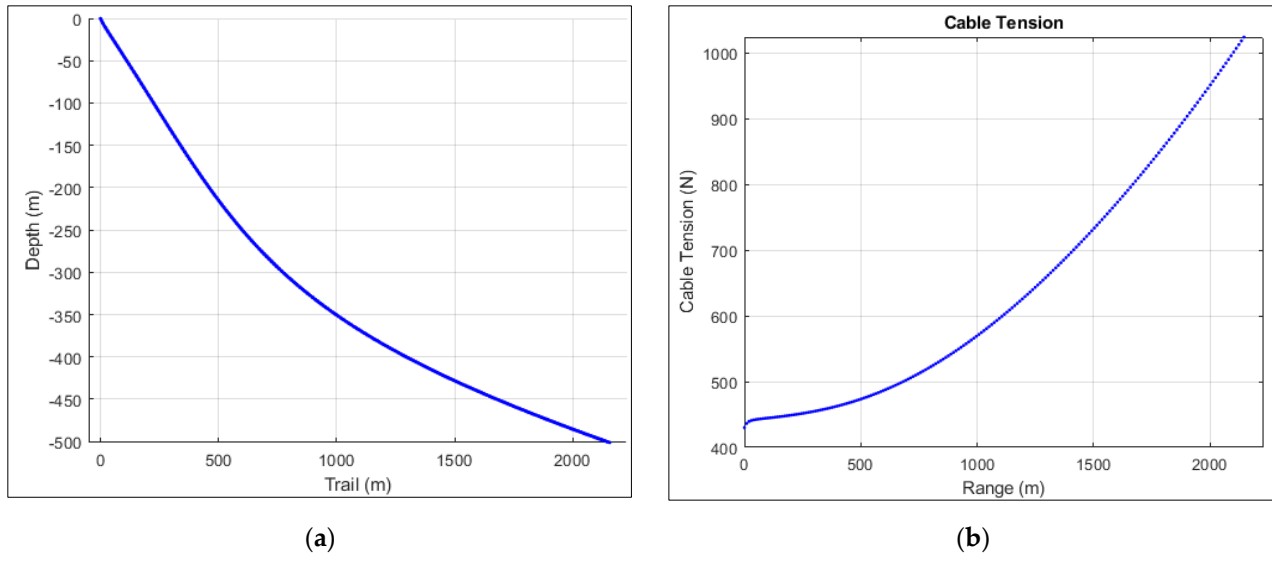

**Figure 15.** TC profile and tension estimated for constant density at surface current 5 Knots: (**a**) TC profile and (**b**) TC Tension.

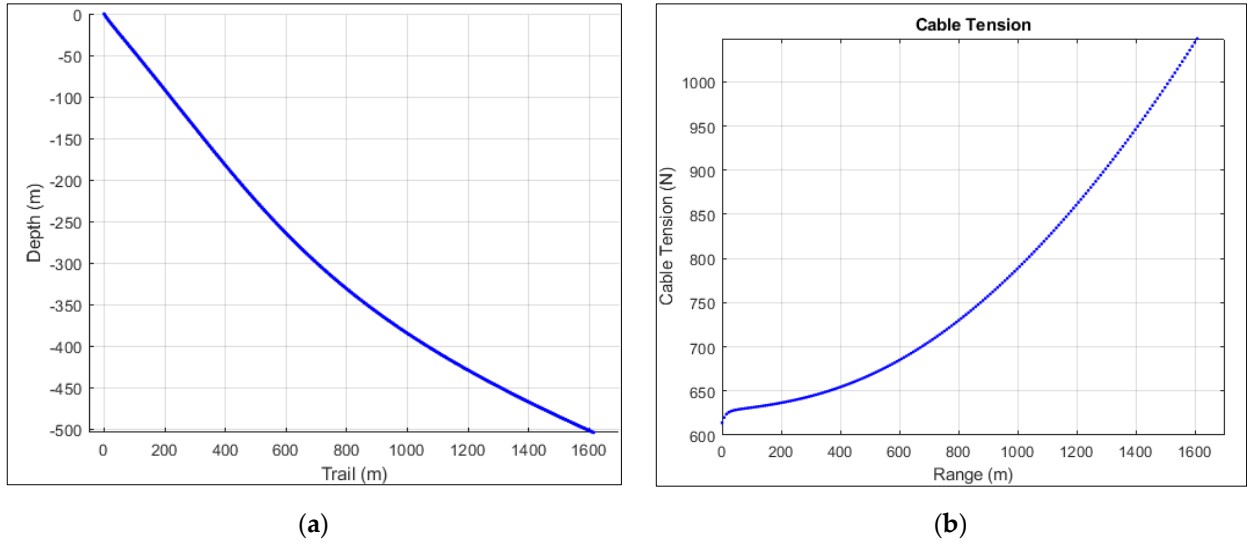

**Figure 16.** TC profile and tension estimated for linear density at surface current 5 Knots: (**a**) TC profile and (**b**) TC Tension.

**Table 3.** Summary of the effect of the density profile on TC profile and tension.

| S. No | Water Density Profile (kg/m$^3$) | TC Density (kg/m$^3$) | Tension (N) | Length of Cable (m) |
|---|---|---|---|---|
| 1 | Constant Profile 1025 | 1050 | 1025 | 2250 |
| 2 | Linear Profile (1025–980) | 1050 | 1040 | 1700 |

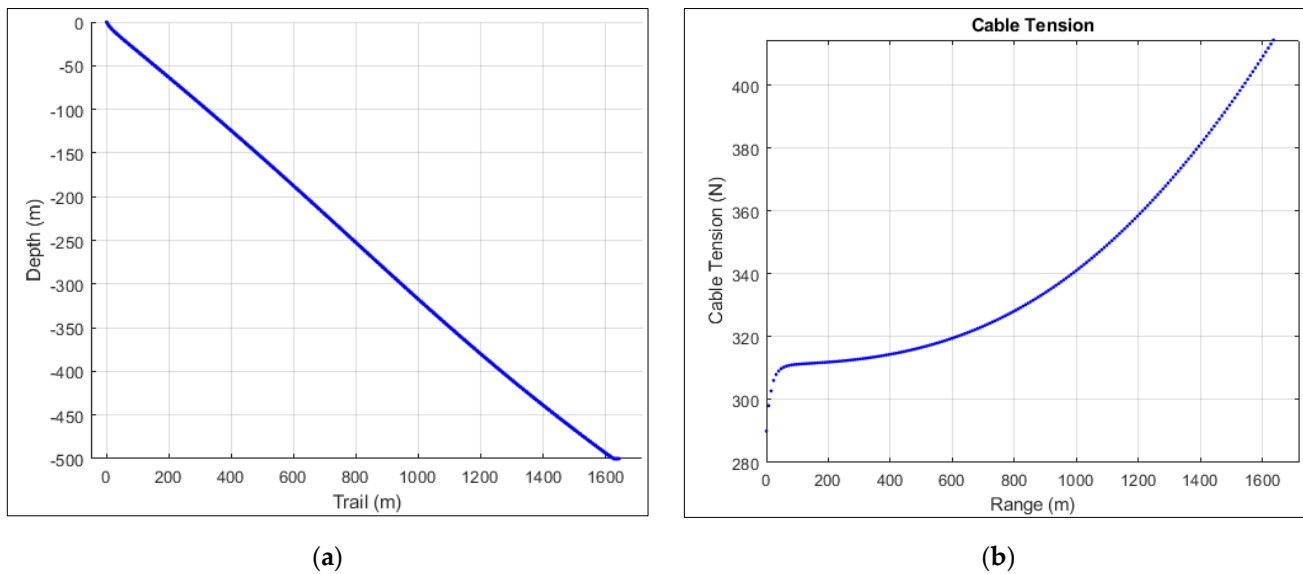

**Figure 17.** TC profile and tension estimated for UMSWR speed 1Knot and surface current 5 Knot: (**a**) TC profile and (**b**) TC Tension.

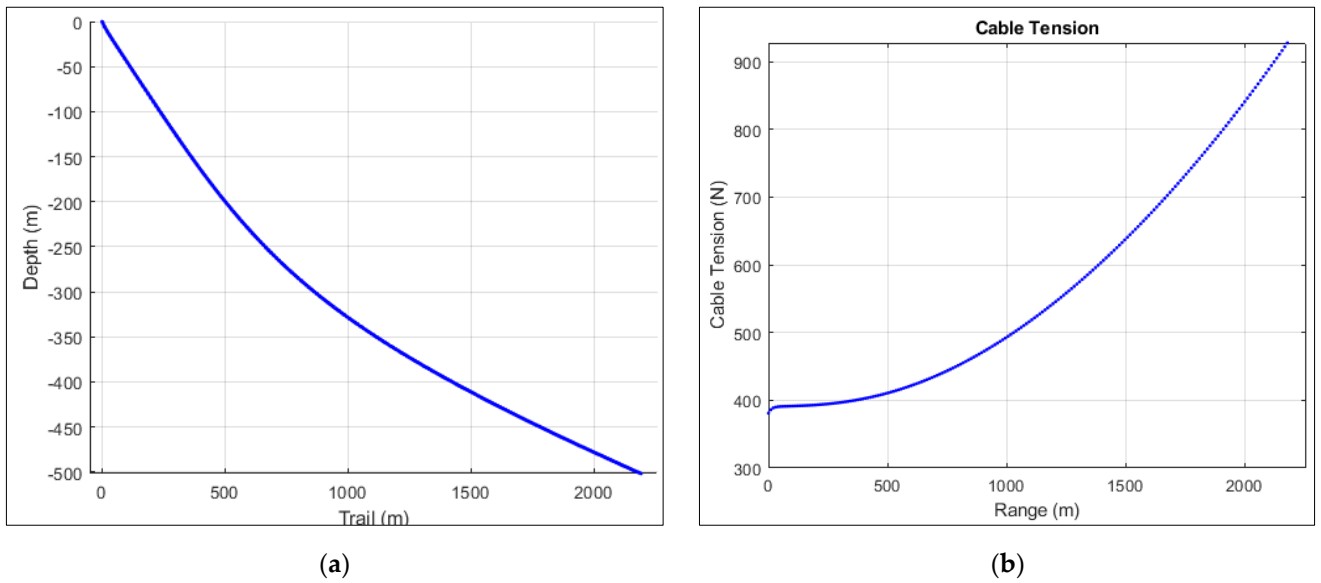

**Figure 18.** TC profile and tension estimated for UMSWR speed 2 Knot, surface current 5 Knot: (**a**) TC profile and (**b**) TC Tension.

**Table 4.** Summary of the effect of the UMSWR speed on TC profile and Tension.

| S. No | UMSWR Speed (Knots) | TC Length Deployed (m) | Tension (N) |
|-------|---------------------|------------------------|-------------|
| 1 | 1 | 1720 | 415 |
| 2 | 2 | 2260 | 928 |

*4.6. Cable Diameter Effects on TC Dynamics*

The effects of TC diameter on tether cable profile and tension are presented in Figures 19–21 for surface current values of 5 Knots and UMSWR speed of 2 Knots at a water depth of 500 m on the seabed. Cable density is taken as $\rho = 1025 \text{ kg/m}^3$. Three cases are considered, 25 mm, 20 mm, and 18 mm cable diameters, respectively. The effects of the simulation are summarized in Table 5. It is obvious from the simulation results that

cable diameter has a big effect on the profile of TC and tension developed in it. Cable diameter is directly related to tension in the cable; the greater the diameter, the greater the tension and vice versa. It is due to the fact that the greater the diameter of the cable, more drag force is developed in the cable, as drag force is directly proportional to the projected area of the cable. Hence, more tension forces are developed in the cable with a bigger diameter. An inverse relation exists for TC diameter with TC length required for a given depth. The greater the diameter of the cable, the lesser will be the length of TC required to arrive at a given depth and vice versa. It is due to the fact that cables with smaller diameters are lighter and, for a given depth of water, takes a larger length to arrive at a certain depth, and cables with larger diameters are heavier, arrive quickly and need shorter lengths of TC comparatively.

**Table 5.** Summary of the effect of the Cable Diameter on TC profile and Tension.

| S. No | Cable Diameter (mm) | Length of Cable Deployed (m) | Tension (N) |
|---|---|---|---|
| 1 | 25 | 2100 | 1178 |
| 2 | 20 | 2300 | 942 |
| 3 | 18 | 2380 | 835 |

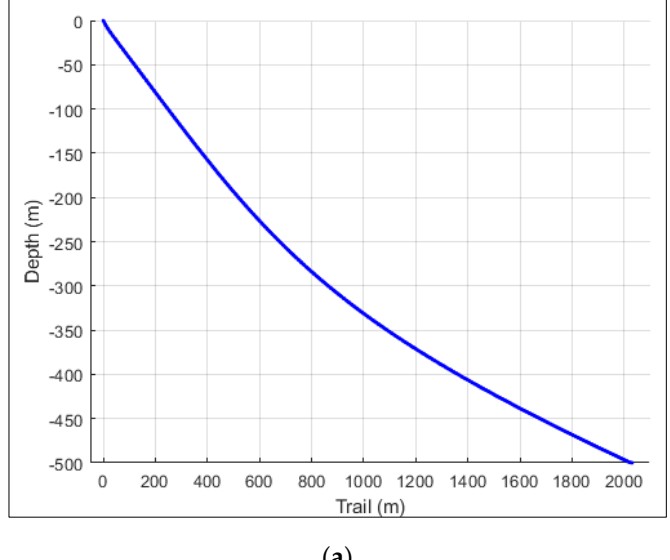

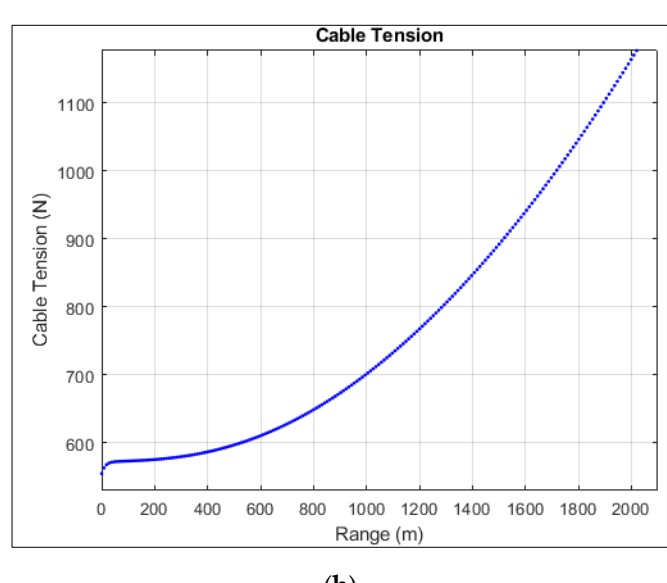

(**a**)        (**b**)

**Figure 19.** TC profile and tension estimated for TC diameter of 25 mm, UMSWR speed of 2 Knot: (**a**) TC profile and (**b**) TC Tension.

### 4.7. Cable Length Changes Effects on TC Dynamics

The parametric study presented here consists of the simulation of TC response to the changes in the TC length due to UMSWR forward speed. This simulation uses a mesh with 200 elements and a time step of 20 s. The effects of UMSWR forward move on TC profile and tension are presented in Figures 22 and 23 for surface current values of 5 Knots, UMSWR speed of 2 Knots and a water depth of 500 m on the seabed. Cable density is taken as $\rho = 1025 \text{ kg/m}^3$. Since the material properties are uniform over the entire scope of TC, the equilibrium profile is expected to be nearly a curve consisting of rectilinear elements inclined to the free surface. During this time, the length of the TC is increasing at a constant rate; TC is subjected to the changing hydrodynamic loads, which drive the cable towards a new equilibrium configuration every moment. To avoid transient cable motions, at the start of the simulation, suitable initial conditions for TC are calculated by executing a separate simulation in which a stationary cable hanging vertically in the water column is carried out. It gives initial dynamic boundary conditions. The simulation is carried out for 3 min in which the initial length of TC increases from 2300 m to 2480 m. The effects of the simulation

are summarized in Table 6. It is obvious from the simulation results that as TC length increases, the tension in the TC increases. This increase in the tension is due to increased drag force on the longer cable. The inverse relation exists for the decrease in TC length.

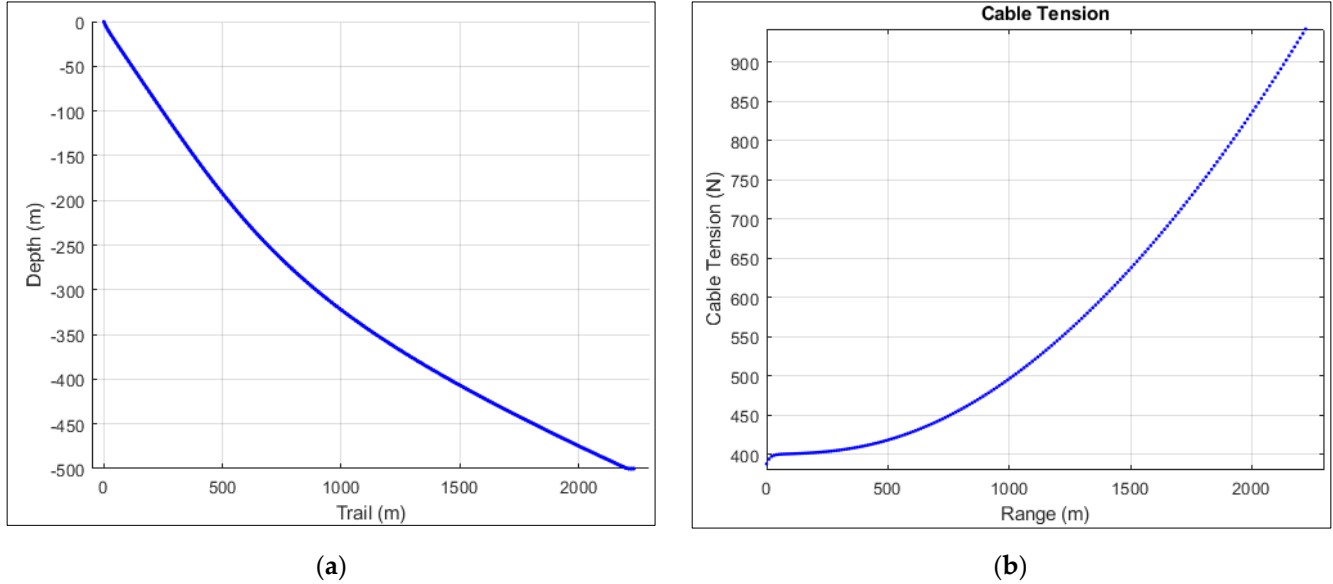

**Figure 20.** TC profile and tension estimated for TC diameter of 20 mm, UMSWR speed 2 Knot: (**a**) TC profile and (**b**) TC Tension.

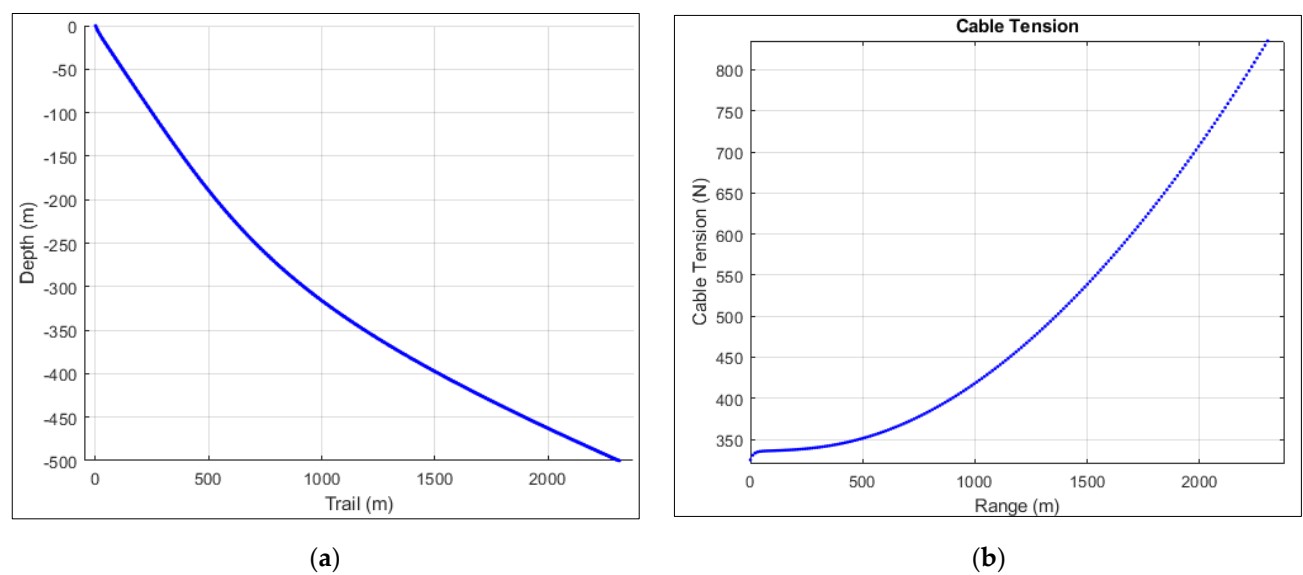

**Figure 21.** TC profile and tension estimated for TC diameter of 20 mm, UMSWR speed 2 Knot: (**a**) TC profile and (**b**) TC Tension.

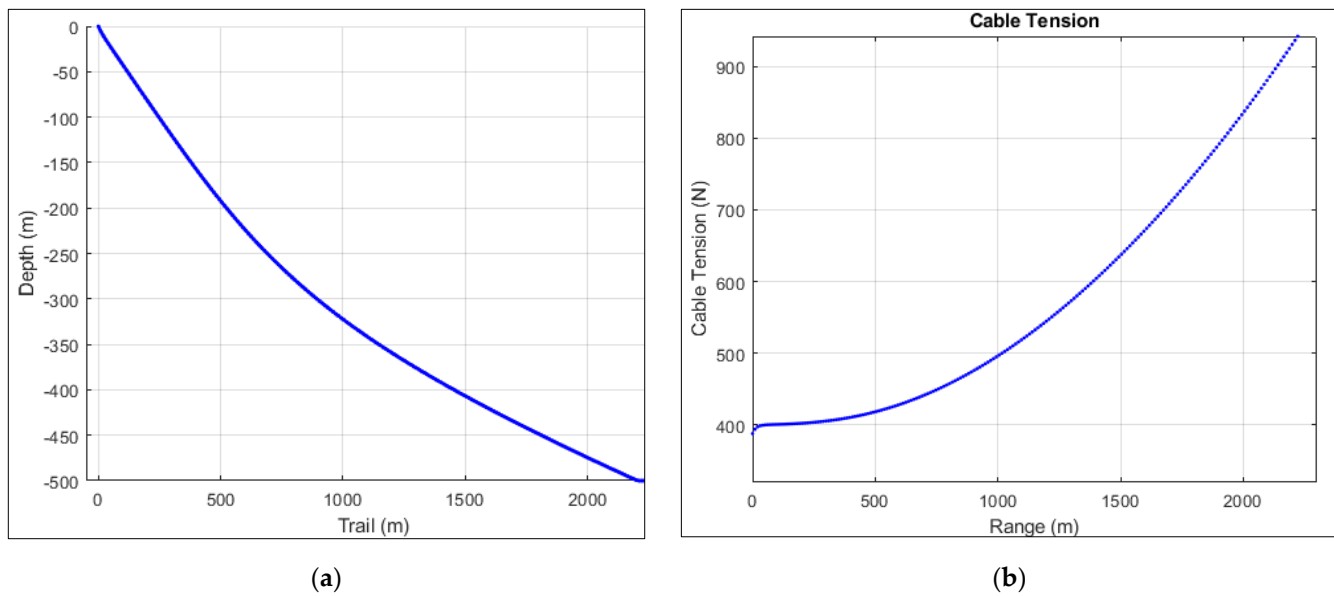

**Figure 22.** TC profile and tension estimate for TC Length 2300 m, depth 500 m: (**a**) TC profile and (**b**) TC Tension.

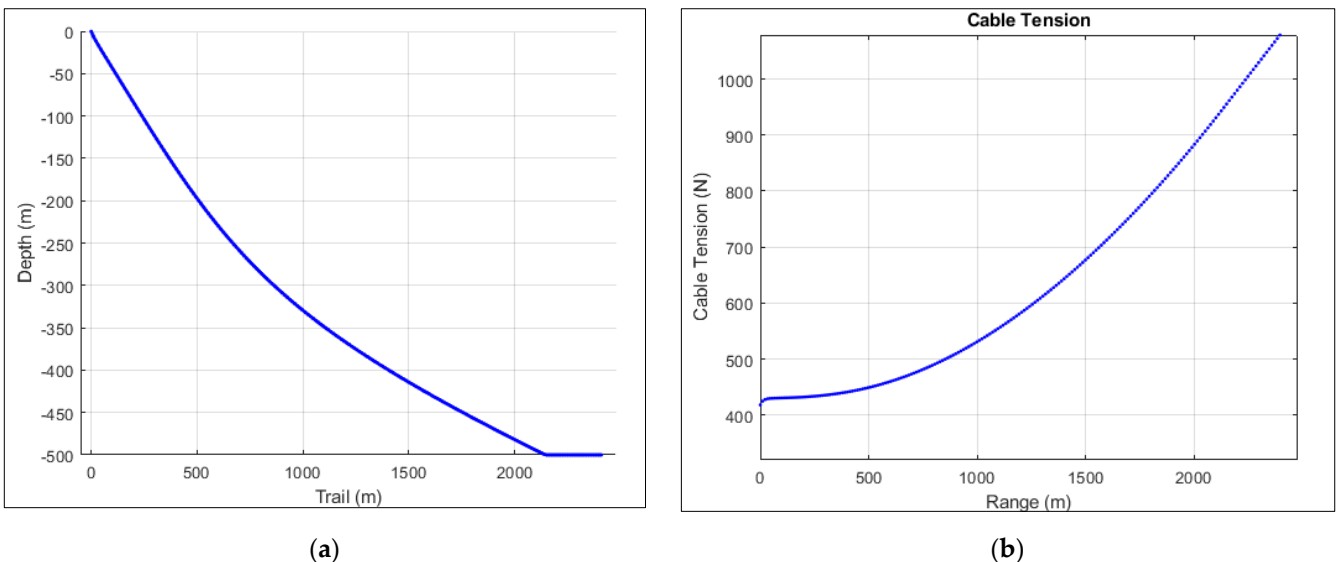

**Figure 23.** TC profile and tension estimate for TC Length 2480 m, depth 500 m: (**a**) TC profile and (**b**) TC Tension.

**Table 6.** Summary of the effect of the TC Length due to UMSWR forward speed of 2 Knots.

| S. No | Initial Length (m) | Segment Length (m) | Tension (N) |
|-------|--------------------|--------------------|-------------|
| 1 | 2300 | 11.5 | 942 |
| 2 | 2320 | 11.6 | 958 |
| 3 | 2340 | 11.7 | 975 |
| 4 | 2360 | 11.8 | 990 |
| 5 | 2380 | 11.9 | 1004 |
| 6 | 2400 | 12 | 1020 |
| 7 | 2420 | 12.1 | 1034 |
| 8 | 2440 | 12.2 | 1050 |
| 9 | 2460 | 12.3 | 1064 |
| 10 | 2480 | 12.4 | 1078 |

### 4.8. Cable Density Effects on TC Dynamics

Regarding the study of the influence of the TC weight/density on TC dynamics, two cases are considered. These are cables of density $\rho = 1025 \text{ kg/m}^3$ and $\rho = 1050 \text{ kg/m}^3$ for the same diameter of the cable. UMSWR speed is 2 Knots, surface water current is 5 Knots, the cable diameter is 20 mm and the depth of the water is 500 m. The effects of TC density on TC profile and tension are presented in Figures 24 and 25. The effects of the simulation are summarized in Table 7. It is obvious from the simulation results that density of the TC has a direct relation with TC tension and inverse relation with the length of the TC for a given depth of UMSWR operation. It is due to the fact that cables with higher density values have greater specific weights and, hence, greater resultant initial tension in the cable, which result in more cable tension at the tow point. Hence, the greater the density of the cable, the greater the tension that is caused, and a lesser length of TC is required for the given depth of UMSWR.

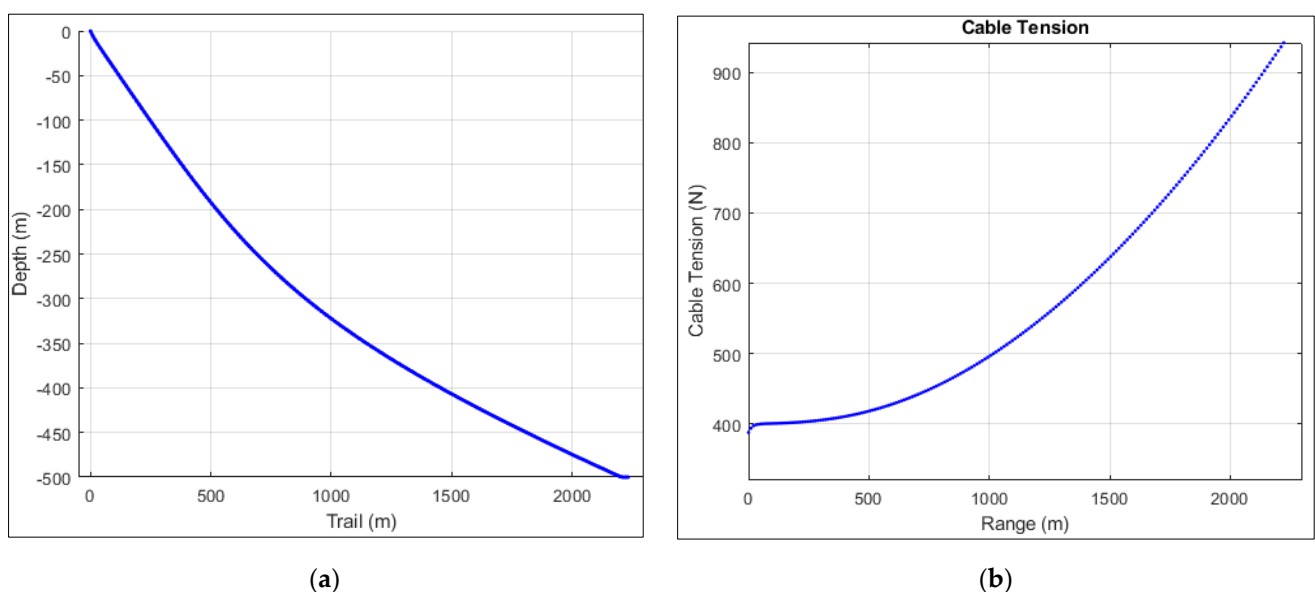

(**a**)            (**b**)

**Figure 24.** TC profile and tension estimate for TC density of 1025 kg/m$^3$: (**a**) TC profile and (**b**) TC Tension.

**Table 7.** Summary of the effect of the TC Density on TC profile and tension.

| S. No | Cable Density $\rho$ (kg/m$^3$) | Length of Cable Deployed (m) | Tension (N) |
|-------|------------------|------------------|------------|
| 1 | 1025 | 2300 | 942 |
| 2 | 1050 | 1700 | 1046 |

### 4.9. Water Depth Effects on TC Dynamics

Finally, the influence of the water depth on TC dynamics is investigated by simulating the cable's response in both shallow and deep water. Shallow water is assumed to be 200 m deep, and deep water is taken as 500 m deep. UMSWR speed is 2 Knots, the surface water current is 5 Knots, the cable diameter is 20 mm, and the cable density is $\rho = 1025 \text{ kg/m}^3$. Five cases of water depth are considered. These are 550 m, 500 m, 200 m, 180 m and 150 m of water depth. Exponential water current profile and linear density profile are considered in the study. The effects of water depth on TC profile and tension are presented in Figures 26–30 for corresponding water depth. The effects of the simulation are summarized in Table 8. It is obvious from simulation results that for shallow water, TCs are easily disturbed by surface currents under slack conditions, therefore, need proportionally longer TCs than those used in deeper waters. Secondly, in shallower water, most of the tension is carried at the TC extremities, making the TC profile concave downward. In

deeper water, the tension is not changing more rapidly but rather changes smoothly, making the TC profile concave upward.

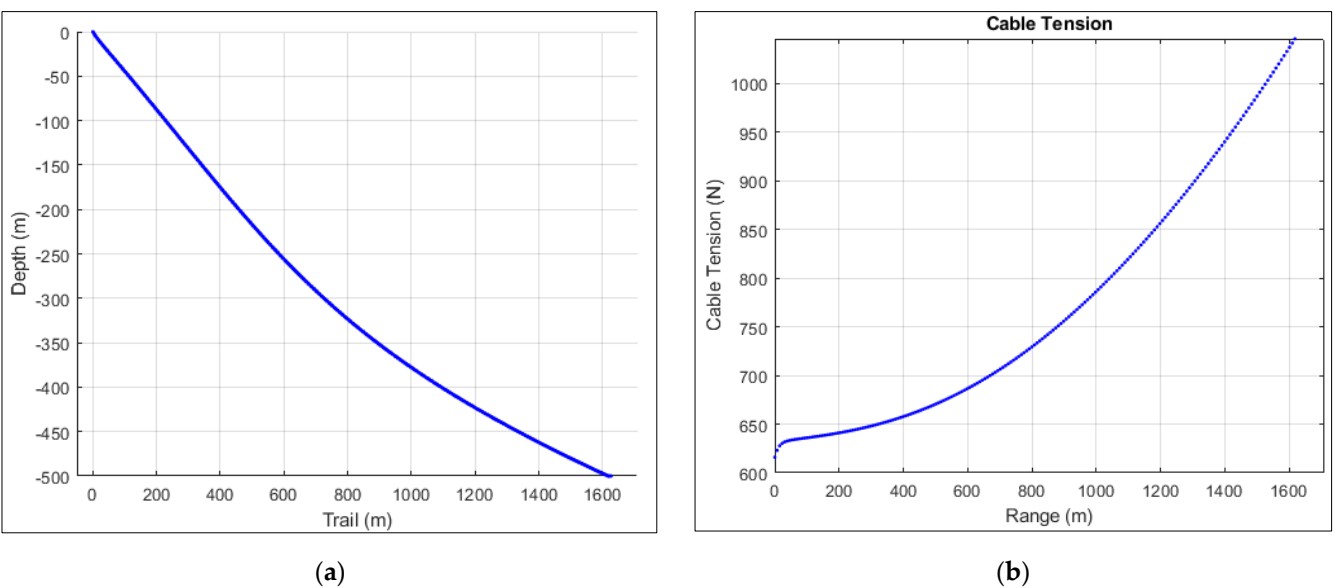

**Figure 25.** TC profile and tension estimate for TC density of 1050 kg/m³: (**a**) TC profile and (**b**) TC Tension.

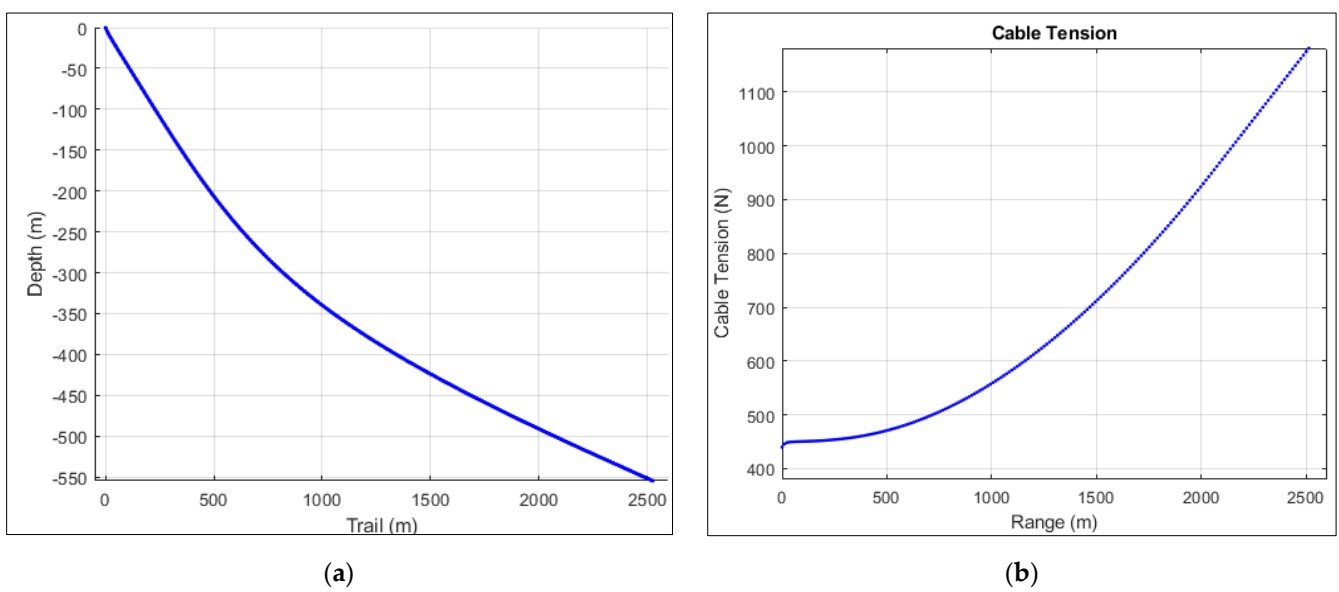

**Figure 26.** TC profile and tension estimate for water depth 550 m UMSWR speed 2 Knots: (**a**) TC profile and (**b**) TC Tension.

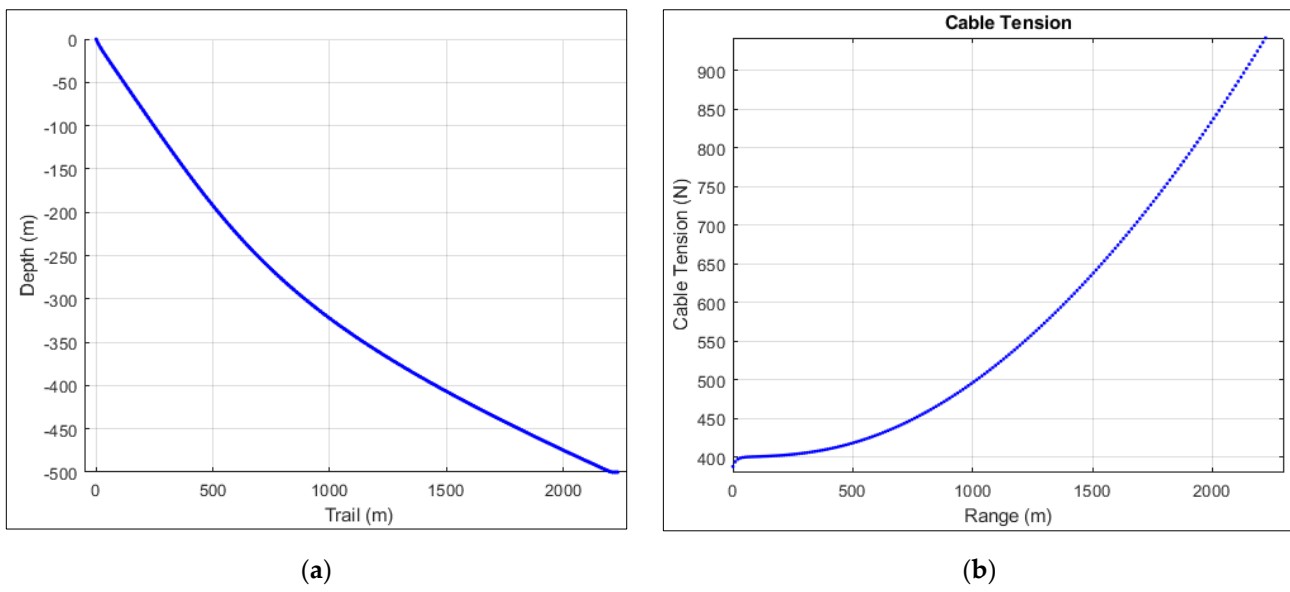

(**a**)                                                                                              (**b**)

**Figure 27.** TC profile and tension estimate for water depth 500 m UMSWR speed 2 Knots: (**a**) TC profile and (**b**) TC Tension.

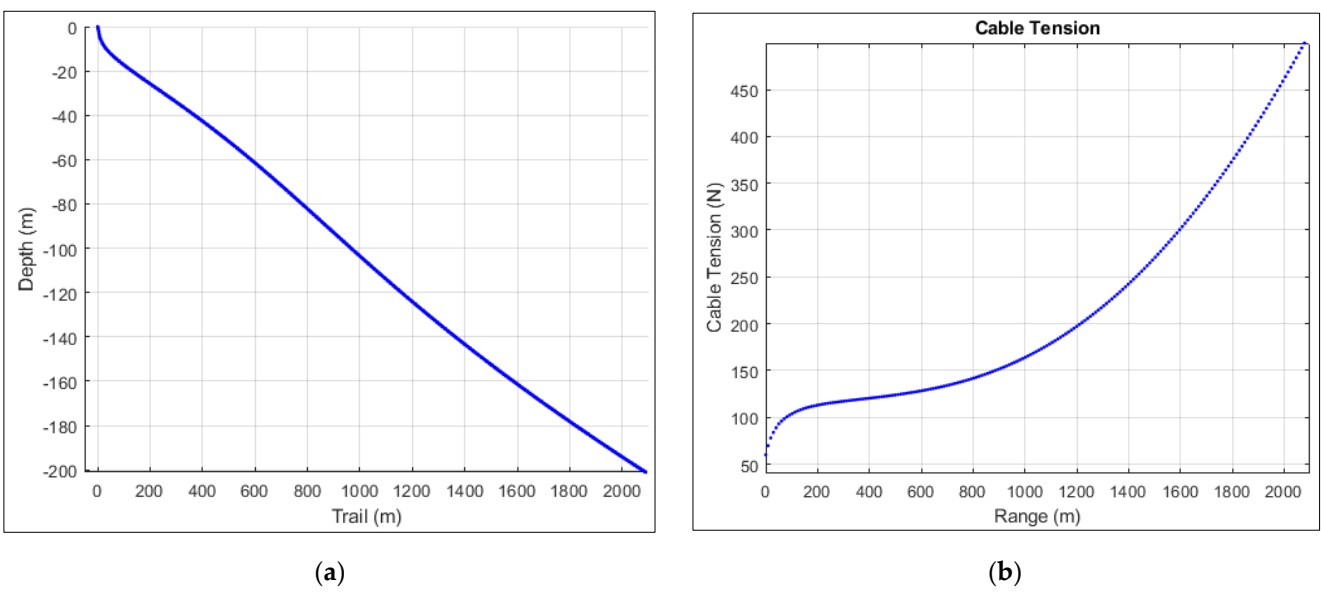

(**a**)                                                                                              (**b**)

**Figure 28.** TC profile and tension estimate for water depth 200 m UMSWR speed 2 Knots: (**a**) TC profile and (**b**) TC Tension.

**Table 8.** Summary of the effect of the TC Density on TC profile and tension.

| S. No | Water Depth (m) | Length of Cable Deployed (m) | Tension (N) |
|-------|-----------------|------------------------------|-------------|
| 1 | 550 | 2600 | 1182 |
| 2 | 500 | 2300 | 942 |
| 3 | 200 | 2100 | 501 |
| 4 | 180 | 1960 | 400 |
| 5 | 150 | 1750 | 295 |

The simulation results revealed that the TC can significantly affect the motion of UMSWR in all cases. While moving on the seabed, the maximum tension in the tether cable is generated at the tow point. This tension causes additional drag forces on the UMSWR.

It is obvious that the simulation results may provide useful guidance and reference for real TC-UMSWR systems in design and operation. The TC configuration can be optimized via numerical simulations, and consequently, TC disturbances on the UMSWR can be reduced. The simulations could also be very helpful in determining the stability and power requirement of UMSWR.

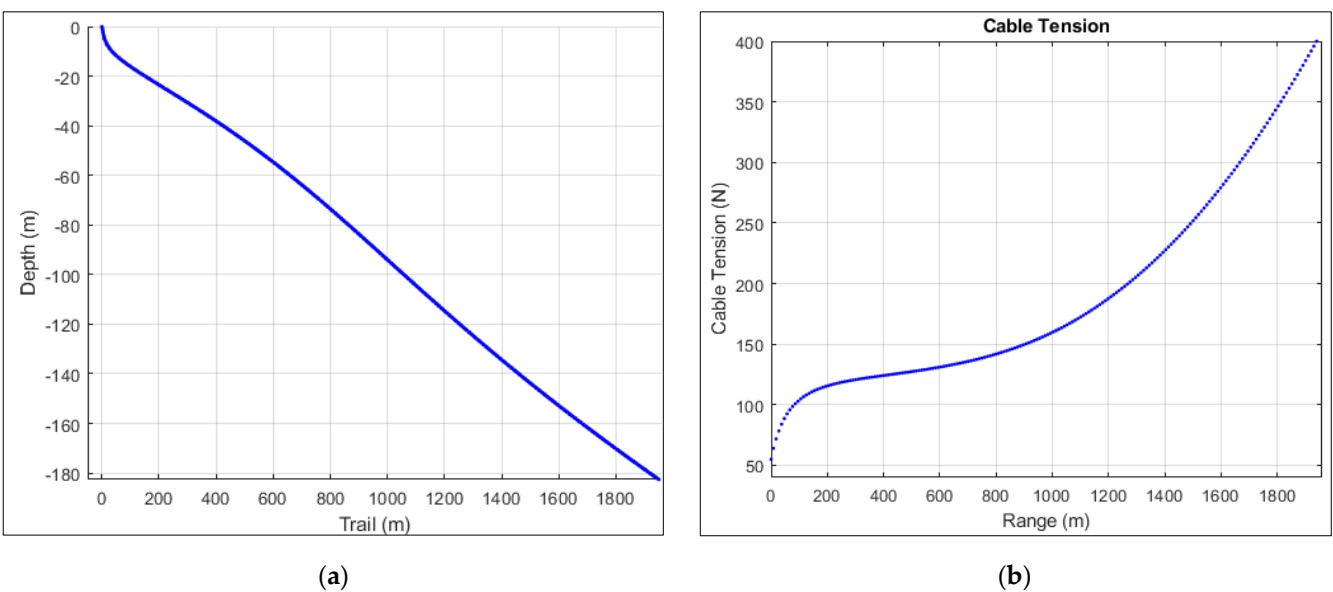

(**a**)                                                                (**b**)

**Figure 29.** TC profile and tension estimate for water depth 180 m UMSWR speed 2 Knots: (**a**) TC profile and (**b**) TC Tension.

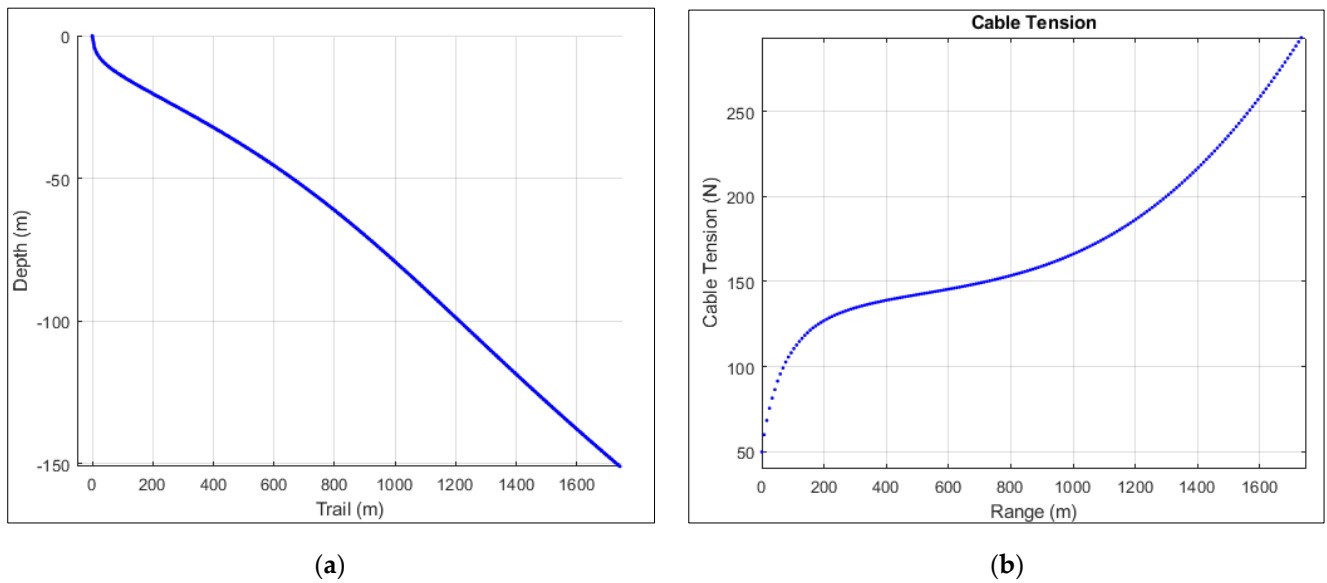

(**a**)                                                                (**b**)

**Figure 30.** TC profile and tension estimate for water depth 150 m UMSWR speed 2 Knots: (**a**) TC profile and (**b**) TC Tension.

## 5. Conclusions

A comprehensive variable-length lumped parameter finite-segment method is developed for tether cable attached to a seabed walking robot. This procedure is intended for the objective of accurate, simple, and efficient simulation of the TC dynamics to determine the profile and tension of TC in the steady state. TC is represented by a series of inextensible rectilinear segments joined by pin joints with the mass of the elements evenly distributed

along the segments. The weight, buoyancy and all external forces are lumped at the center of the mass of the element. The effects of environmental parameters of variable water current and variable water density on TC dynamics are included. These environment parameters are specified in a mathematically precise way and included in the model without introducing unnecessary growth in the dimensions or complexity of the final model. This modeling is believed to be both accurate and efficient and, at the same time, comprehensive and applicable to a broad range of both fixed-length and variable-length underwater TC systems. A computer code developed in MATLAB software has been implemented, and several numerical simulation examples have been worked out, showing almost the exact performance of the method. Various aspects and parameters of the system are investigated, thereby demonstrating the versatility of the method. The simulation results revealed that the TC tension causes additional drag forces on the UMSWR and can significantly affect the motion of UMSWR in all cases. Moreover, the simulation results may provide useful guidance and reference for real TC-UMSWR systems in design and operation.

The presented work will promote the understanding of the physical mechanisms within the underwater tether cable that influence the motion of the seabed walking robot. The main contribution of our work is summarized here:

i. A mathematical analysis method is proposed for calculating the position, posture, and tension force in underwater tether cable for seabed walking robots. Lumped parameter and finite-segment method is used based on the Morrison equation in the steady state.

ii. The developed mathematical model is simulated numerically using codes in MATLAB software. The influences of various input parameters on output values in the mathematical analysis and numerical simulation are evaluated and discussed.

iii. Moreover, the influences of the environmental parameters (variable water density and variable surface current) are included in both the mathematical model as well in numerical simulation.

From both academic and practical points of view, this approach is applicable in many instances of underwater research and exploration operations. It can be applied to both low-tension and high-tension steady-state underwater tethered systems of ocean monitoring and measurement applications, ROVs, Towed cable-body systems and tethered seabed walking vehicles for underwater operations. Future work can be extended by considering the motion of the TC -UMSWR coupled system for various input parameters, with verification via water tests.

**Author Contributions:** Conceptualization, A.K., L.W. and X.W.; methodology, A.K. and X.W.; software, A.K. and A.E.; validation, A.K., X.W. and Z.L.; formal analysis, A.K. and Z.L.; investigation, A.K. and X.W.; resources, X.W. and L.W.; data curation, A.K., M.I. and Z.L.; writing, original draft preparation, A.K. and X.W.; writing, review and editing, A.K. and M.I.; visualization, A.K. and Z.L.; supervision, L.W. and X.W.; project administration, L.W.; funding acquisition, L.W. All authors have read and agreed to the published version of the manuscript.

**Funding:** This research was funded by the National Natural Science Foundation of China, grant number 51779064.

**Institutional Review Board Statement:** Not applicable.

**Informed Consent Statement:** Not applicable.

**Data Availability Statement:** This study is a new application to the existing methods of underwater cable dynamics. A novel approach has been adapted for including the variable environmental parameters in a mathematically precise way.

**Conflicts of Interest:** The authors declare no conflict of interest.

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
