# Peer review of "Analytical and Numerical Study of Underwater Tether Cable Dynamics for Seabed Walking Robots Using Quasi-Static Approximation"

_jmse, doi:10.3390/jmse11081539_

Round 1

Reviewer 1 Report

In this article was described a method to analyse the effect generated by cable on a legged robot on seabed. The experimentation was carried on using software and it was validated through many examples.

Some comments for authors:

1. The support of the cable on ship is a wincher. However, the dynamical effect generated by wincher when the cable is tensioned is not taken in account because the description of the problem describes the analysis just in equilibrium condition. Why is the reason?

2. The situation where the solution was analyse is when the robot on seabed is in contact with floor. But the dynamical behaviour when the robot is sumerged in any fluid is lost.

3. The cable structure is mandatory for modelling. Maybe the location for this figure and their descriptioncould be better in next section called Modelling of the tehet cable system.

4. In figure 5, the representation of each reference frame is confused because is used a n index. I understand that the n index means each frame is evaluated in a sequential procedure to representate dynamic behaviour whole cable. Could you do more clear?

5. The equation used for showing the convertion between the system refers to fixed frame uses a rotational matrix but in manuscript is not visualized correctly. Please check it the nomenclature.

6. The assumptions which ones give inputs to this method limit the approach to reach a functional dynamical behaviour. I propose to carry on with others assumptions otherwise with less asumptions. For example, increasing to 6DOF the model.

7. I have experience with this issue in so many experiments carried out in real enviroments. I could say that the tether effect could be modeled as in different ways depends of approach. Could you compare this method in real environments?

Reviewer 2 Report

The paper performs an analytical and numerical study on the dynamics of the underwater mooring cable for the seabed walking robot using quasi-static approximation. In general, all parts of the work are well done.

As suggestions for improvement of the paper I suggest the authors to describe in the conclusion section, the usefulness/novelty of the mathematical model and to detail the benefits of using such an approach proposed by the paper in practice.

I also recommend checking the text for language.

I recommend accepting the paper, and making the small corrections suggested.

Reviewer 3 Report

The paper presents an interesting numerical study on underwater tether cable dynamics for seabed walking robot using quasi-static approximation. The paper in general is well written and comprehensive. It seems to provide appropriate up-to-date descriptions of the most well-known methods applied to numerical modelling applied in similar systems. The topic seems to be original. The proposed method is investigated and verified sufficiently with MATLAB simulation examples. The conclusions address sufficiently the main research question posed and are consistent with the evidence and arguments presented. The references are sufficient and appropriate. Overall the paper seems to be of sufficiently good quality.

Follows a list of a few minor comments in order to help the authors to improve the paper quality.

There are a few grammatical errors, particularly in the Abstract (e.g., iput, ouput).

Also, on lines 14 and 116, the word “Waking” must be corrected (to “Walking”).

There are several cases where the term “modeling” is spelled either with the American (modeling) or the British (modelling) spelling. The authors must choose only one case to use (perhaps for the journal is preferred the American spelling).

The term “steady-state”, is also sometimes written as “steady-state” and in other cases as “steady state”. Again the authors must choose only one case to use (perhaps “steady-state”).

Although in sections 1.Introduction and 2.Problem Statement the motivations and aims are presented clearly, I would suggest to summarize in a paragraph as well the contributions too.

On line 226, I think the equation should be numbered (as (3)). However, perhaps this is not very convenient since must be changed the numbering of all the following equations, as well as the references to them within the text. Therefore, I do not insist.

The references to the tables within the text use the word “table” with a small first letter (e.g., table 1, table, 2, etc..). I think the word “table” should start with a capital letter (i.e. Table 1, Table 2, …).

There is also a misuse of the word “Figure”. In some cases the figures are referred as “Figure …”, in other cases as “Fig …”, and in some cases as “fig …”. Please use only one option (perhaps “Figure …” is the accepted term for the journal).

The quality of English language in general is fine.

Reviewer 4 Report

Overall, the manuscript does fit the scope of JMSE. In my opinion, it is a study to deal with a necessary problem, however, there are a lot of things that need to be addressed to meet the quality publication. Some of the concerns are as follows:

1/ The writing format of the article needs to be amended, in particular, all the text should follow the standard of the journal in font size. Refine your paper and do neat formatting of your paper.

2/ Despite the motivating topic, the theoretical contribution of the manuscript does not seem significant, the explanation of the procedure and contents of the paper are ambiguous and the experimental contribution is minor since the manuscript does not present results from real trials. The novelty of the approach is rather low since it seems all the methods are existing. This paper should have more theoretical contributions.

3/ The motivation and background of wide practical use of the theoretic results presented should be clearly emphasized to facilitate the readers. What are the underlying factors that led to better performance of the proposed method?

4/ Write the organization of the paper in the introduction part.

5/ In the introduction part, the literature survey is quite good. However, I think that the authors could enrich the reference section by discussing some new works related to the latest publications on the mooring system or cable system applied to marine platforms that refer to the discussed subject. As far as I know, there are several methods with different assumptions and considerations for the motion of cable or mooring line which include analytical method, Lumped mass method, catenary method and Finite Difference method. Also, the authors should give a detailed explanation about the main merit and shortcoming of the above method modeling for the mooring lines or cable system. To help the authors in this direction, I suggest the following reference: study on dynamic behavior of unmanned surface vehicle-linked unmanned underwater vehicle system for underwater exploration, a study on hovering motion of the underwater vehicle with umbilical cable, study on the dynamic behaviors of an USV with a ROV. Moreover, it is suggested that the novel index of this paper should be explained in detail. And the introduction should be added to do a better job of explaining the existing methods and why they are or are not valuable. The objective/problem statement need to be explained. What are the novelties of the proposed method? what the research challenges/motivations of the paper are?

6/ The author should emphasize the difference between the current model developed and the research methods of past scholars? Why does the author’s discussion of this topic produce academic or practical value? That is, the reference value of research contributions and practical feasibility must be rigorously stated and  in order to present the value of research topics.

7/ The question of validating the results of numerical simulations is missing. Or links to works where such validation is done. The results of laboratory experiments or full-scale measurements.

8/ In simulation part, more design parameters and comparisons with some existing results are recommended to prove the efficacy of the proposed method. The authors should compare the proposed approach with other techniques to testify the effectiveness of the proposed strategy. I would like to mention about highlighted by the author’s effectiveness of the presented method. If expected achievement of the method application was its better efficiency, than I would suggest presentation of some comparison with other effective method. A comparison between the results of this paper and the related ones in the literature can be done by items. So, the contribution of this paper can be clear.

9/ The explanations and analysis of some simulation results should be enriched to show the validity of data.

10/ The comment about the mooring modelling and the effects of mooring forces on the motions of the platform are missing in the paper. This needs to be incorporated. The authors need to study the effect of cable mooring forces on the motions of the system in the simulation results?

11/ Detailed implementation information should be provided (hardware, software, configuration, settings). A detailed discussion of hardware and software applied to your system should be mentioned. Provide specifications of the hardware and software used for simulation of the approach.

12/ From the simulations with satisfactory results, the system performance is expected in actual experiments with your proposed method. If a maritime test is not available, an appropriate analysis software could act as an alternative.

13/ Some typos and grammatical errors should be checked carefully, and some formatting problems that need to be revised carefully.

Some typos and grammatical errors should be checked carefully, and some formatting problems that need to be revised carefully.

Round 2

Reviewer 1 Report

The suggestions were added/corrected in manuscript

Reviewer 4 Report

Thank you for the revised manuscript. In a general way, most of my comments were answered by the authors. I have no further comments.

Ok!